# The generation of cortical novelty responses through inhibitory plasticity

**Auguste Schulz[1,2†], Christoph Miehl[1,3†], Michael J Berry II[4], Julijana Gjorgjieva[1,3*]**

[1]Max Planck Institute for Brain Research, Frankfurt, Germany; [2]Technical University of Munich, Department of Electrical and Computer Engineering, Munich, Germany; [3]Technical University of Munich, School of Life Sciences, Freising, Germany; [4]Princeton University, Princeton Neuroscience Institute, Princeton, United States

**Abstract** Animals depend on fast and reliable detection of novel stimuli in their environment. Neurons in multiple sensory areas respond more strongly to novel in comparison to familiar stimuli. Yet, it remains unclear which circuit, cellular, and synaptic mechanisms underlie those responses. Here, we show that spike-timing-dependent plasticity of inhibitory-to-excitatory synapses generates novelty responses in a recurrent spiking network model. Inhibitory plasticity increases the inhibition onto excitatory neurons tuned to familiar stimuli, while inhibition for novel stimuli remains low, leading to a network novelty response. The generation of novelty responses does not depend on the periodicity but rather on the distribution of presented stimuli. By including tuning of inhibitory neurons, the network further captures stimulus-specific adaptation. Finally, we suggest that disinhibition can control the amplification of novelty responses. Therefore, inhibitory plasticity provides a flexible, biologically plausible mechanism to detect the novelty of bottom-up stimuli, enabling us to make experimentally testable predictions.

**\*For correspondence:**
gjorgjieva@brain.mpg.de

[†]These authors contributed equally to this work

**Competing interests:** The authors declare that no competing interests exist.

## Introduction

In an ever-changing environment, animals must rapidly extract behaviorally useful information from sensory stimuli. Appropriate behavioral adjustments to unexpected changes in stimulus statistics are fundamental for the survival of an animal. We still do not fully understand how the brain detects such changes reliably and quickly. Local neural circuits perform computations on incoming sensory stimuli in an efficient manner by maximizing transmitted information or minimizing metabolic cost (*Simoncelli and Olshausen, 2001*; *Barlow, 2013*). Repeated or predictable stimuli do not provide new meaningful information. As a consequence, one should expect that responses to repeated stimuli are suppressed – a phenomenon postulated by the framework of predictive coding (*Clark, 2013*; *Spratling, 2017*). Recent experiments have demonstrated that sensory circuits across different modalities can encode a sequence or expectation violation and can detect novelty (*Keller et al., 2012*; *Natan et al., 2015*; *Zmarz and Keller, 2016*; *Hamm and Yuste, 2016*; *Homann et al., 2017*). The underlying neuronal and circuit mechanisms behind expectation violation and novelty detection, however, remain elusive.

A prominent paradigm used experimentally involves two types of stimuli, the repeated (or frequent) and the novel (or deviant) stimulus (*Näätänen et al., 1982*; *Fairhall, 2014*; *Natan et al., 2015*; *Homann et al., 2017*; *Weber et al., 2019*). Here, the neuronal responses to repeated stimuli decrease, a phenomenon that is often referred to as adaptation (*Fairhall, 2014*). Adaptation can occur over a wide range of timescales, which range from milliseconds to seconds (*Ulanovsky et al., 2004*; *Lundstrom et al., 2010*), and to multiple days in the case of behavioral habituation (*Haak et al., 2014*; *Ramaswami, 2014*). We refer to the elevated neuronal response to a novel stimulus, compared to the response to a repeated stimulus, as a 'novelty response' (*Homann et al., 2017*). Responses to repeated versus novel stimuli, more generally, have also been studied on

different spatial scales spanning the single neuron level, cortical microcircuits and whole brain regions. At the scale of whole brain regions, a widely studied phenomenon is the mismatch negativity (MMN), which is classically detected in electroencephalography (EEG) data and often based on an auditory or visual 'oddball' paradigm (*Näätänen et al., 1982*; *Hamm and Yuste, 2016*). The occasional presentation of the so-called oddball stimulus among frequently repeated stimuli leads to a negative deflection in the EEG signal – the MMN (*Näätänen et al., 2007*).

Experiments at the cellular level typically follow the oddball paradigm with two stimuli that, if presented in isolation, would drive a neuron equally strongly. However, when one stimulus is presented frequently and the other rarely, the deviant produces a stronger response relative to the frequent stimulus (*Ulanovsky et al., 2003*; *Nelken, 2014*; *Natan et al., 2015*). The observed reduction in response to the repeated, but not the deviant, stimulus has been termed stimulus-specific adaptation (SSA) and has been suggested to contribute to the MMN (*Ulanovsky et al., 2003*). SSA has been observed in multiple brain areas, most commonly reported in the primary auditory cortex (*Ulanovsky et al., 2003*; *Yaron et al., 2012*; *Natan et al., 2015*; *Seay et al., 2020*) and the primary visual cortex (*Movshon and Lennie, 1979*; *Hamm and Yuste, 2016*; *Vinken et al., 2017*; *Homann et al., 2017*). Along the visual pathway, SSA has also been found at different earlier stages including the retina (*Schwartz et al., 2007*; *Geffen et al., 2007*; *Schwartz and Berry, 2008*) and the visual thalamic nuclei (*Dhruv and Carandini, 2014*; *King et al., 2016*).

To unravel the link between multiple spatial and temporal scales of adaptation, a variety of mechanisms has been proposed. Most notably, modeling studies have explored the role of adaptive currents, which reduce the excitability of the neuron (*Brette and Gerstner, 2005*), and short-term depression of excitatory feedforward synapses (*Tsodyks et al., 1998*). Most models of SSA in primary sensory areas of the cortex focus on short-term plasticity and the depression of thalamocortical feedforward synapses (*Mill et al., 2011a*; *Mill et al., 2011b*; *Park and Geffen, 2020*). The contribution of other mechanisms has been under-explored in this context. Recent experimental studies suggest that inhibition and the plasticity of inhibitory synapses shape the responses to repeated and novel stimuli (*Chen et al., 2015*; *Kato et al., 2015*; *Natan et al., 2015*; *Hamm and Yuste, 2016*; *Natan et al., 2017*; *Heintz et al., 2020*). Natan and colleagues observed that in the mouse auditory cortex, both parvalbumin-positive (PV) and somatostatin-positive (SOM) interneurons contribute to SSA (*Natan et al., 2015*). Furthermore, neurons that are more strongly adapted receive stronger inhibitory input than less adapted neurons, suggesting potentiation of inhibitory synapses as an underlying mechanism (*Natan et al., 2017*). In the context of habituation, inhibitory plasticity has been previously hypothesized to be the driving mechanism behind the reduction of neural responses to repeated stimuli (*Ramaswami, 2014*; *Barron et al., 2017*). Habituated behavior in *Drosophila*, for example, results from prolonged activation of an odor-specific excitatory subnetwork, which leads to the selective strengthening of inhibitory synapses onto the excitatory subnetwork (*Das et al., 2011*; *Glanzman, 2011*; *Ramaswami, 2014*; *Barron et al., 2017*).

Here, we focus on the role of inhibitory spike-timing-dependent plasticity (iSTDP) in characterizing neuronal responses to repeated and novel stimuli at the circuit level. We base our study on a recurrent spiking neural network model of the mammalian cortex with biologically inspired plasticity mechanisms that can generate assemblies in connectivity and attractors in activity to represent the stimulus-specific activation of specific sub-circuits (*Litwin-Kumar and Doiron, 2014*; *Zenke et al., 2015*; *Wu et al., 2020*). We model excitatory and inhibitory neurons and include stimulus-specific input not only to the excitatory but also to the inhibitory population, as found experimentally (*Ma et al., 2010*; *Griffen and Maffei, 2014*; *Znamenskiy et al., 2018*). This additional assumption readily leads to the formation of specific inhibitory-to-excitatory connections through inhibitory plasticity (*Vogels et al., 2011*), as suggested by recent experiments (*Lee et al., 2014*; *Xue et al., 2014*; *Znamenskiy et al., 2018*; *Najafi et al., 2020*).

We demonstrate that this model network can generate excess population activity when novel stimuli are presented as violations of repeated stimulus sequences. Our framework identifies plasticity of inhibitory synapses as a sufficient mechanism to explain population novelty responses and adaptive phenomena on multiple timescales. In addition, stimulus-specific inhibitory connectivity supports adaptation to specific stimuli (SSA). This finding reveals that the network configuration encompasses computational capabilities beyond those of intrinsic adaptation. Furthermore, we suggest disinhibition to be a powerful regulator of the amplification of novelty responses. Our modeling framework enables us to formulate additional experimentally testable predictions. Most intriguing,

we hypothesize that neurons in primary sensory cortex may not signal the violation of periodicity of a sequence based on bottom-up input, but rather adapt to the distribution of presented stimuli.

## Results

### A recurrent neural network model with plastic inhibition can generate novelty responses

Recent experimental studies have indicated an essential role of inhibitory circuits and inhibitory plasticity in adaptive phenomena and novelty responses (*Chen et al., 2015*; *Kato et al., 2015*; *Natan et al., 2015*; *Hamm and Yuste, 2016*; *Natan et al., 2017*; *Heintz et al., 2020*). To understand if and how plastic inhibitory circuits could explain the emergence of novelty responses, we built a biologically plausible spiking neuronal network model of recurrently connected 4000 excitatory and 1000 inhibitory neurons based on recent experimental findings on tuning, connectivity, and inhibitory and excitatory STDP in the cortex (Materials and methods). Excitatory-to-excitatory (E-to-E) synapses were plastic based on the triplet spike-timing-dependent plasticity (eSTDP) rule (*Sjöström et al., 2001*; *Pfister and Gerstner, 2006*; *Gjorgjieva et al., 2011*; *Figure 1—figure supplement 1*). The triplet STDP rule enabled the formation of strong bidirectional connections among similarly selective neurons (*Gjorgjieva et al., 2011*; *Montangie et al., 2020*). Plasticity of connections from inhibitory to excitatory neurons was based on an inhibitory STDP (iSTDP) rule measured experimentally (*D'amour and Froemke, 2015*), and shown to stabilize excitatory firing rate dynamics in recurrent networks (*Vogels et al., 2011*; *Figure 1—figure supplement 1*). In contrast to other frameworks which have found short-term plasticity as key for capturing adaptation phenomena, we only included long-term plasticity and did not explicitly model additional adaptation mechanisms.

We targeted different subsets of excitatory and inhibitory neurons with different external stimuli, to model that these neurons are stimulus-specific ('tuned') to a given stimulus (*Figure 1A*, left, see Materials and methods). One neuron could be driven by multiple stimuli. Starting from an initially randomly connected network, presenting tuned input led to the emergence of excitatory assemblies, which are strongly connected, functionally related subsets of excitatory neurons (*Figure 1—figure supplement 2C*, left). Furthermore, tuned input also led to the stimulus-specific potentiation of inhibitory-to-excitatory connections (*Figure 1—figure supplement 2E*, left). We refer to this part of structure formation as the 'pretraining phase' of our simulations (Materials and methods). This pretraining phase imprinted structure in the network prior to the actual stimulation paradigm as a model of the activity-dependent refinement of structured connectivity during early postnatal development (*Thompson et al., 2017*).

To test the influence of inhibitory plasticity on the emergence of a novelty response, we followed an experimental paradigm used to study novelty responses in layer 2/3 (L2/3) of mouse primary visual cortex (V1) (*Homann et al., 2017*). In *Homann et al., 2017*, a single stimulus consisted of 100 randomly oriented Garbor patches. Three different stimuli (A, B, and C) were presented in a sequence (ABC) (*Figure 1A*, right). The same sequence (ABC) was then repeated several times in a sequence block. In the second-to-last sequence, the last stimulus was replaced by a novel stimulus (N). In the consecutive sequence block, a new sequence with different stimuli was presented (we refer to this as a unique sequence stimulation paradigm). The novel stimuli were also different for each sequence block. In this paradigm, we observed elevated population activity in the excitatory model population at the beginning of each sequence block ('onset response') and a steady reduction to a baseline activity level for the repeated sequence presentation (*Figure 1B*). Upon presenting a novel stimulus, the excitatory population showed excess activity, clearly discernible from baseline, called the 'novelty response'. This novelty response was comparable in strength to the onset response. Sorting spike rasters according to sequence stimuli revealed that stimulation leads to high firing rates in the neurons that are selective to the presented stimulus (A, B, or C) (*Figure 1C*). When we used a different set of stimuli in the stimulation versus the pretraining phase to better match the randomly oriented Gabor patches presented in *Homann et al., 2017* (*Figure 1—figure supplement 3A*, see Materials and methods), we found the same type of responses to repeated and novel stimuli (*Figure 1—figure supplement 3B*). When examining a random subset of neurons, we found general response sparseness and periodicity during sequence repetitions (*Figure 1D*), very

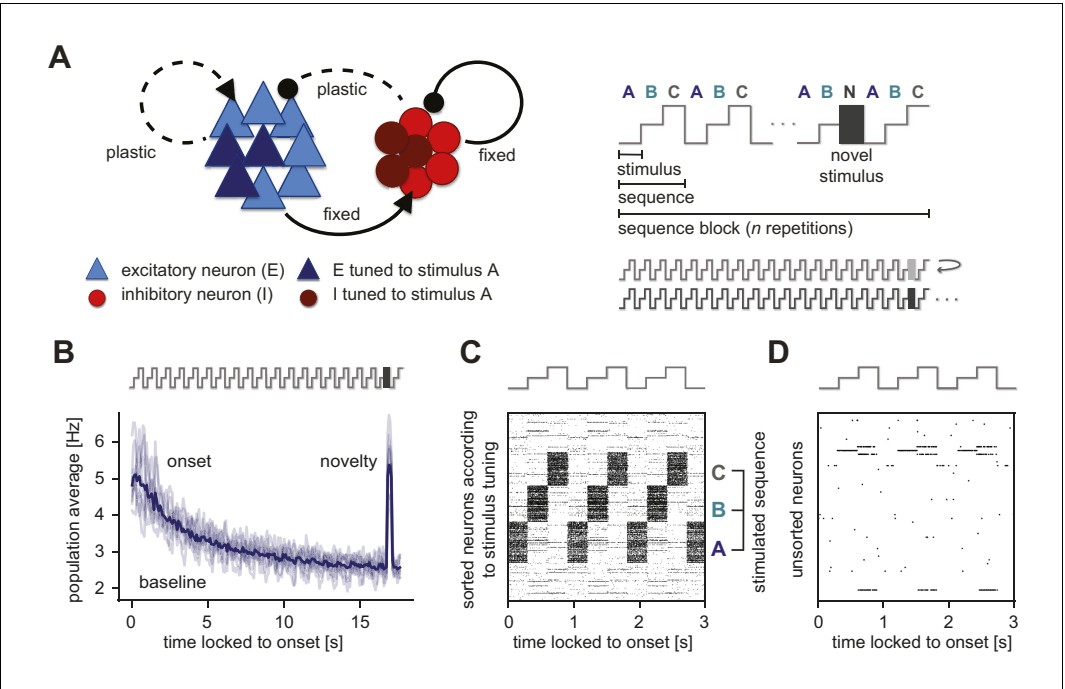

**Figure 1.** Generation of novelty responses in a recurrent plastic neural network model. (**A**) Left: A recurrently connected network of excitatory (E) neurons (blue triangles) and inhibitory (I) neurons (red circles) receiving tuned input. Excitatory neurons tuned to a sample stimulus A are highlighted in dark blue, the inhibitory counterparts in dark red. E-to-E synapses and I-to-E synapses were plastic, and all other synapses were fixed. Right: Schematic of the stimulation protocol. Multiple stimuli (A, B, and C) were presented in a sequence (ABC). Each sequence was repeated *n* times in a sequence block. In the second-to-last sequence, the last stimulus was replaced by a novel stimulus (N). Multiple sequence blocks followed each other without interruption, with each block containing sequences of different stimuli. (**B**) Population average firing rate of all excitatory neurons as a function of time after the onset of a sequence block. Activity was averaged (solid line) across multiple non-repeated sequence blocks (transparent lines: individual blocks). A novel stimulus (dark gray) was presented as the last stimulus of the second-to-last sequence. (**C**) Spiking activity in response to a sequence (ABC) in a subset of 1000 excitatory neurons where the neurons were sorted according to the stimulus from which they receive tuned input. A neuron can receive input from multiple stimuli and can appear more than once in this raster plot. (**D**) A random unsorted subset of 50 excitatory neurons from panel C. Time was locked to the sequence block onset.

The online version of this article includes the following figure supplement(s) for figure 1:

**Figure supplement 1.** Excitatory and inhibitory synaptic plasticity functions for different pairing frequencies.
**Figure supplement 2.** Strong connections form between excitatory and excitatory, as well as inhibitory and excitatory neuron groups that are tuned to the same stimulus.
**Figure supplement 3.** Different stimuli in the pretraining and stimulation phases generate similar synaptic weight and firing rate dynamics.
**Figure supplement 4.** Quantifying response density in the unique sequence stimulation paradigm.
**Figure supplement 5.** Normalization time step $\Delta t$ does not affect the occurrence of a novelty response.

similar to experimental findings (*Homann et al., 2017*). More concretely, sparse population activity for repeated stimuli in our model network was the result of each stimulus presentation activating a subset of excitatory neurons in the network, which were balanced by strong inhibitory feedback. Therefore, only neurons that directly received this feedforward drive were highly active, while most other neurons in the network were instead rather silent. Periodicity in the activity of single neurons resulted from the repetition of a sequence.

In the model, the fraction of active excitatory neurons was qualitatively similar for novel, adapted and onset stimuli (*Figure 1—figure supplement 4*). The relatively sparse novelty response in our model was the result of increased inhibition onto all excitatory neurons in the network, with activity remaining mainly in the neurons tuned to the novel stimulus. In contrast, *Homann et al., 2017* found that a large fraction of neurons respond to a novel stimulus, suggesting a dense novelty response.

Since the increase in inhibition seems to be responsible for the absence of a dense novelty response in our model, in a later section we suggest disinhibition as a mechanism to achieve the experimentally observed dense novelty responses in our model.

Our results suggest that presenting repeated stimuli (and repeated sequences of stimuli) to a plastic recurrent network with tuned excitatory and inhibitory neurons readily leads to a reduction of the excitatory averaged population response, consistent with the observed adaptation in multiple experimental studies in various animal models and brain regions (*Ulanovsky et al., 2003*; *Hamm and Yuste, 2016*; *Homann et al., 2017*). Importantly, the model network generates a novelty response when presenting a novel stimulus by increasing the excitatory population firing rate at the time of stimulus presentation (*Näätänen et al., 2007*).

## The dynamics of novelty and onset responses depend on sequence properties

To explore the dynamics of novelty responses, we probed the model network with a modified stimulation paradigm. Rather than fixing the number of sequence repetitions in one sequence block (*Figure 1A*, right), here we presented a random number of sequence repetitions (nine values between 4 and 45 repetitions) for each sequence block. This allowed us to measure the novelty and onset responses as a function of the number of sequence repetitions. Novelty and onset responses were observed after as few as four sequence repetitions (*Figure 2A*). After more than 15 sequence repetitions, the averaged excitatory population activity reached a clear baseline activity level (*Figure 2A*). The novelty response amplitude, measured by the population rate of the novelty peak minus the baseline population rate, increased with the number of sequence repetitions before saturating for a high number of sequence repeats (*Figure 2B*, black dots). The onset response amplitude after the respective sequence block followed the same trend (*Figure 2B*, gray dots). Next, we varied the number of stimuli in a sequence, resulting in different sequence lengths across blocks (3 to 15 stimuli per sequence). By averaging excitatory population responses across sequence blocks with equal length, we found that the decay of the onset response depends on the number of stimuli in a sequence (*Figure 2C*). Upon fitting an exponentially decaying function to the activity of the onset response, we derived a linear relationship between the number of stimuli in a sequence and the decay constant (*Figure 2D*).

In summary, we found that novelty responses arise for different sequence variations. Our model network suggests that certain features of the novelty response depend on the properties of the presented sequences. Changing the number of sequence repetitions modifies the onset and novelty response amplitude (*Figure 2A,B*), while a longer sequence length leads to a longer adaptation time constant (*Figure 2C,D*). Interestingly, both findings are in good qualitative agreement with experimental data that presented similar sequence variations (*Homann et al., 2017*). An exponential fit of the experimental data found a time constant of $\tau = 3.2 \pm 0.7$ repetitions when the number of sequence repetitions was varied (*Homann et al., 2017*). The time constant in our model network was somewhat longer ($\tau = 9 \pm 1$ repetitions), but on a similar order of magnitude (*Figure 2B*). Similarly, our model network produced a linear relationship between the adaptation time constant and sequence length with a slope of $m = 1.6 \pm 0.04$ (*Figure 2D*), very close to the slope extracted from the data ($m = 2.1 \pm 0.3$) (*Homann et al., 2017*). Therefore, grounded on biologically-plausible plasticity mechanisms, and capable of capturing the emergence and dynamics of novelty responses, our model network provides a suitable framework for a mechanistic dissection of the circuit contributions in the generation of a novelty response.

## Stimulus periodicity in the sequence is not required for the generation of a novelty response

Experimental studies have often reported novelty or deviant responses by averaging across several trials due to poor signal-to-noise ratios of the measured physiological activity (*Homann et al., 2017*; *Vinken et al., 2017*). Therefore, we investigated the network response to paradigms with repeated individual sequence blocks (*Figure 3A*), which we refer to as the repeated sequence stimulation paradigm. We randomized the order of the sequence block presentation to avoid additional temporal structure beyond the stimulus composition of the sequences. Repeating sequence blocks dampened the onset response at sequence onset compared to the unique sequence stimulation paradigm

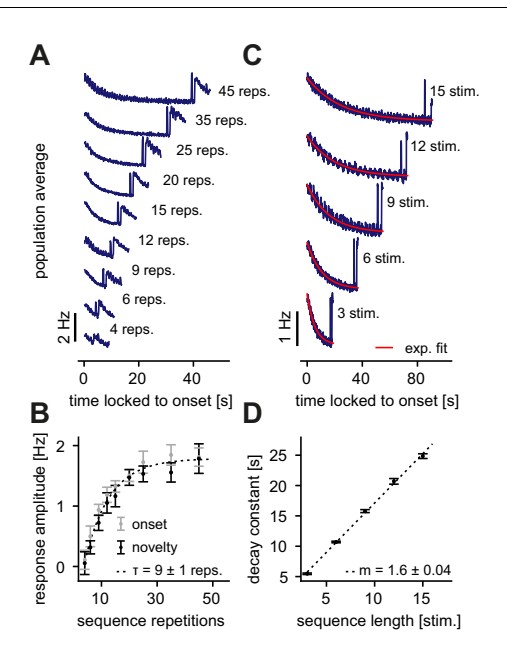

**Figure 2.** Dependence of the novelty response on the number of sequence repetitions and the sequence length. (**A**) Population average firing rate of all excitatory neurons for a different number of sequence repetitions within a sequence block. Time is locked to the sequence block onset. (**B**) The response amplitude of the onset (gray) and the novelty (black) response as a function of sequence repetitions fit with an exponential with a time constant $\tau$. (**C**) Population average firing rate of all excitatory neurons for varying sequence length fit with an exponential function (red). Time is locked to the sequence block onset. (**D**) The onset decay time constant (fit with an exponential, as shown in panel C) as a function of sequence length. The simulated data was fit with a linear function with slope $m$. (**B**, **D**) Error bars correspond to the standard deviation across five simulated instances of the model.

(compare *Figure 1B* and *Figure 2A,B* with *Figure 3A*). Next, we wondered whether the excitatory and inhibitory population responses to repeated and novel stimuli are related. We found that both excitatory and inhibitory populations adapt to the repeated stimuli and show a prominent novelty peak that is larger than the respective averaged onset response (*Figure 3B,C*). Based on these findings, we make the following predictions for future experiments: (1) A novelty response is detectable in both the excitatory and inhibitory populations. (2) The sequence onset response is dampened for multiple presentations of the same sequence block compared to the presentation of unique sequence blocks.

Next, we investigated whether the generation of novelty responses observed in the model network depends on the sequence structure. If the novelty responses were to truly signal the violation of the sequence structure or the stimulus predictability in a sequence, we would expect a novelty response to occur if two stimuli in a sequence were swapped, that is, ACB instead of ABC. We found that swapping the last and second-to-last stimulus, instead of presenting a novel stimulus, does not elicit a novelty response (*Figure 3D*). Additionally, we asked whether the periodicity of the stimuli within a sequence influences the novelty response. Shuffling the stimuli within a sequence block still generates a novelty response and adaptation to the repeated stimuli, similar to the strictly periodic case (*Figure 3E*, compare to *Figure 3A*). Finally, we investigated if the novelty peak depends on the input firing rate of the novel stimulus. We found that a reduction of the input drive decreases the novelty peak, revealing a monotonic dependence of the novelty response on stimulus strength (*Figure 3F*). Based on these results, we make two additional predictions: (3) The periodicity of stimuli in the sequence is not required for the generation of a novelty response. Hence, the novelty response encodes the distribution of presented stimuli, rather than the structure of a sequence. (4) A novelty response depends on the strength of the novel stimulus.

## Increased inhibition onto highly active neurons leads to adaptation

To gain an intuitive understanding for the sensitivity of novelty responses to stimulus identity but lack of sensitivity to stimulus periodicity in the sequence, we more closely examined the role of inhibitory plasticity as the leading mechanism behind the novelty responses in our model. We found that novelty responses arise because inhibitory plasticity fails to sufficiently increase inhibitory input and to counteract the excess excitatory input into excitatory neurons upon the presentation of a novel stimulus. In short, novelty responses can be understood as the absence of adaptation in an otherwise adapted response. Adaptation in the network arises through increased inhibition onto highly active neurons through selective strengthening of I-to-E weights (*Figure 4A*).

To determine how inhibitory plasticity drives the generation of novelty responses or, equivalently, adaptation in our model, we studied the evolution of inhibitory weights. The inhibitory weights onto stimulus-specific assemblies tuned to the stimuli in a given sequence increased upon presentation of

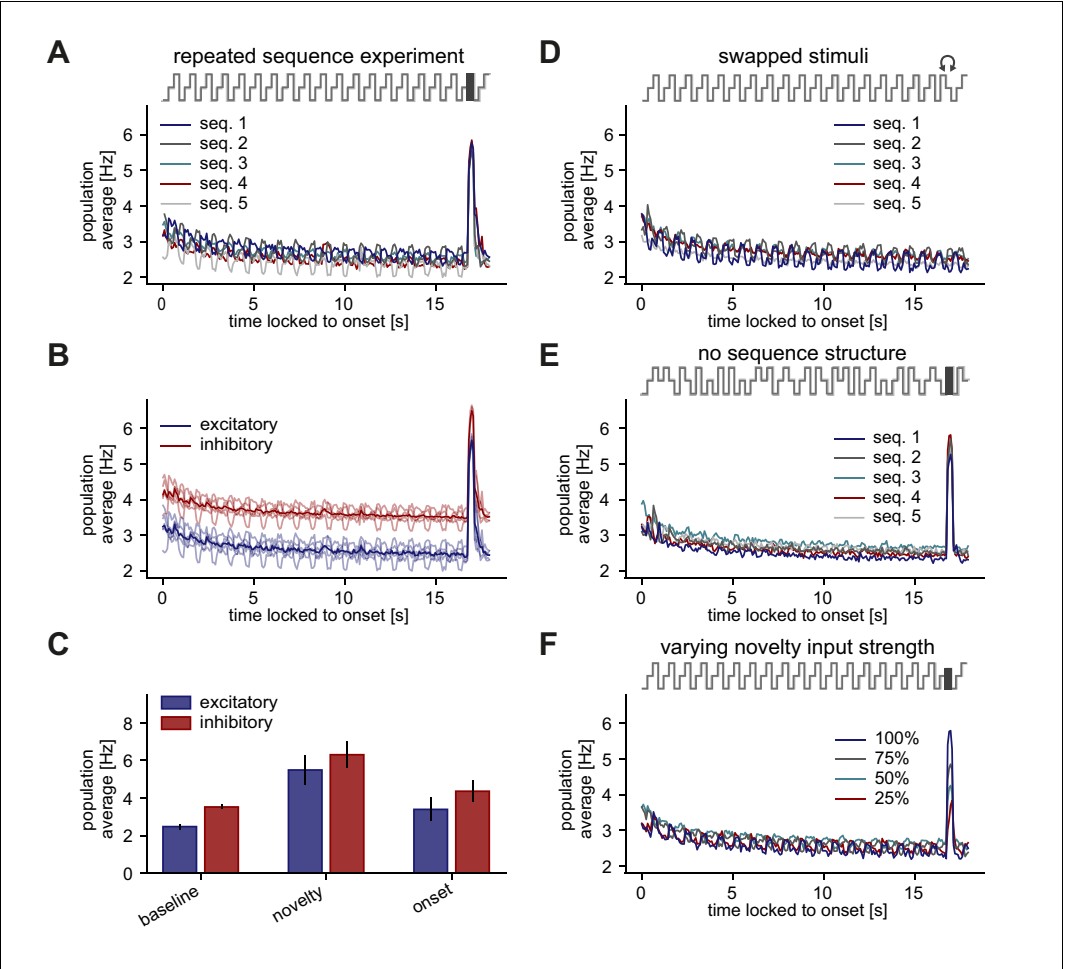

**Figure 3.** Stimulus periodicity in the sequence is not required for the generation of a novelty response. (**A–F**) Population average firing rate of all excitatory neurons (and all inhibitory neurons in B,C) during the presentation of five different repeated sequence blocks. The population firing rate was averaged across ten repetitions of each sequence block. Time is locked to sequence block onset. (**A**) A novel stimulus was presented as the last stimulus of the second-to-last sequence. (**B**) Same as panel A but for both excitatory and inhibitory populations (transparent lines: individual sequence averages). (**C**) Comparison of baseline, novelty, and onset response for inhibitory and excitatory populations. Error bars correspond to the standard deviation across the five sequence block averages shown in B. (**D**) In the second-to-last sequence, the last and second-to-last stimulus were swapped instead of presenting a novel stimulus. (**E**) Within a sequence, stimuli were shuffled in a pseudo-random manner where a stimulus could not be presented twice in a row. A novel stimulus was presented as the last stimulus of the second-to-last sequence. (**F**) A novel stimulus was presented as the last stimulus of the second-to-last sequence. Each sequence had a different feedforward input drive for the novel stimulus, indicated by the percentage of the typical input drive for the novel stimulus used before.

the corresponding sequence block, and decreased otherwise (**Figure 4B**). The population firing rate during repeated presentation of a sequence decreased (adapted) on the same timescale as the increase of the inhibitory weights related to this sequence (**Figure 4C**). When a stimulus was presented to the network for the first time, the total excitatory input to the corresponding excitatory neurons was initially not balanced by inhibition. Hence, the neurons within the assembly tuned to that stimulus exhibited elevated activity at sequence onset, leading to what we called the 'onset response' (**Figure 1B**). The same was true for the novelty responses as reflected in low inhibitory weights onto novelty assemblies relative to repeated assemblies (**Figure 1—figure supplement 2D, E**). Consequently, the generation of a novelty response did not depend on the specific periodicity of the stimuli within a sequence (**Figure 3**). Swapping two stimuli did not generate a novelty response since the corresponding assemblies of each stimulus were already in an adapted state. Therefore,

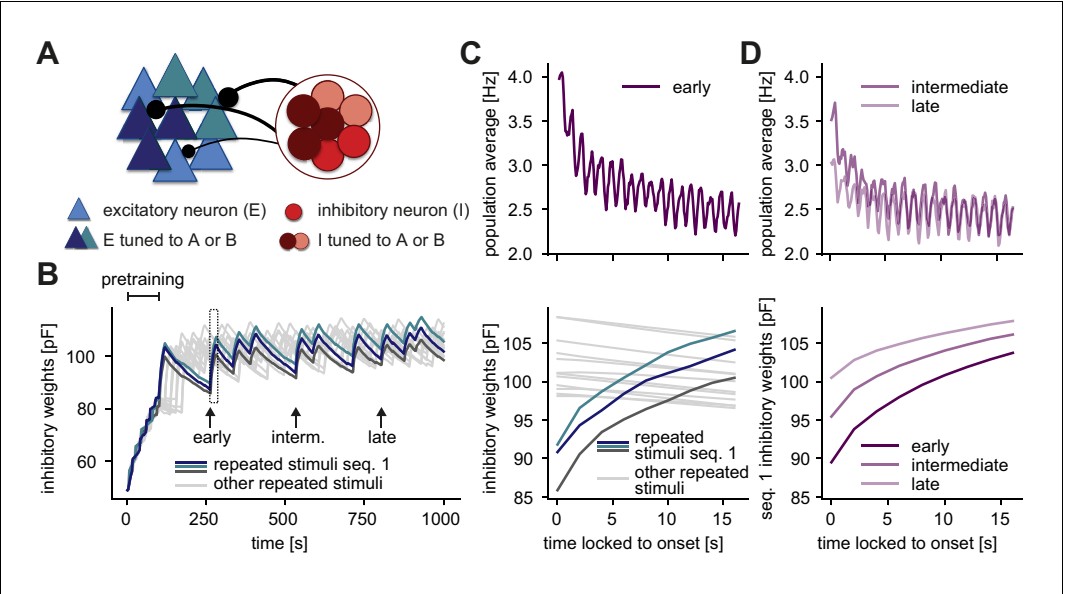

**Figure 4.** Inhibition onto neurons tuned to repeated stimuli increases during sequence repetitions. (**A**) Schematic of increased inhibitory weights onto two stimulus-specific assemblies upon the repeated presentation of stimuli A and B (indicated in dark blue and turquoise) relative to neurons from other assemblies (light blue). (**B**) Evolution of the average inhibitory weights onto stimulus-specific assemblies. Colored traces mark three stimulus-specific assemblies in sequence 1: A, B, and C. Arrows indicate time points of early, intermediate, and late sequence block presentation shown in C and D. (**C**) Top: Population average firing rate of all excitatory neurons during the repeated presentation of sequence 1 at an early time point (see panel B). Time is locked to sequence onset. Bottom: Close-up of panel B (rectangle). Time is locked to sequence onset. (**D**) Top: Same as panel C (top) but at intermediate and late time points (see panel B). Bottom: Corresponding dynamics of the average inhibitory weights onto all three stimulus-specific assemblies from sequence 1 at early, intermediate and late time points (see panel B). The dark purple trace (early) corresponds to the average of the three colored traces in C (bottom). The online version of this article includes the following figure supplement(s) for figure 4:

**Figure supplement 1.** Pretraining parameters do not qualitatively influence the novelty response.
**Figure supplement 2.** Fast inhibitory plasticity is key for the generation of a novelty response.

our results suggest that the exact sequence structure of stimulus presentations is not relevant for the novelty response, as long as the overall distribution of stimuli is maintained.

Interestingly, we found that adaptation occurs on multiple timescales in our model. The fastest is the timescale of milliseconds on which inhibitory plasticity operates, the next slowest is the timescale of seconds corresponding to the presentation of a sequence block, and finally the slowest is the timescale of minutes corresponding to the presentation of the same sequence block multiple times (*Figure 4D*, top; also compare *Figure 1B* and *Figure 3A*). The slowest decrease in the population firing rate was the result of long-lasting changes in the average inhibitory weights onto the excitatory neurons tuned to the stimuli within a given sequence. Hence, the average inhibitory weight for a given sequence increased with the number of previous sequence block presentations of that sequence (*Figure 4D*, bottom).

Using a different set of stimuli in the stimulation versus the pretraining phase to match the randomly oriented Gabor patches presented in *Homann et al., 2017*, led to qualitatively similar firing rate and synaptic weight dynamics (*Figure 1—figure supplement 3C,D*, see also Materials and methods). Differences in the mean inhibitory weights onto different stimulus-specific assemblies in a given sequence were due to random initial differences in assembly size and connection strength (*Figure 4B,C*, see Materials and methods). Differences in early, intermediate, and late inhibitory weight changes, however, were consistent across different experiments and model instantiations (*Figure 4D*, *Figure 1—figure supplement 3D*, right).

Furthermore, we observed that the dynamics of inhibitory plasticity and the generation of a novelty response did not depend on the exact parameters of the pretraining phase (*Figure 4—figure*

*supplement 1*). Specifically, increasing the number of repetitions in the pretraining phase increased the height of the novelty peak, but eventually reached a plateau at 10 repetitions (*Figure 4—figure supplement 1A*). Increasing the number of stimuli decreased the height of the novelty peak (*Figure 4—figure supplement 1C*). However, these pretraining parameters only affected some aspects of the novelty response, but preserved the generation of the novelty response. Even without a pretraining phase (zero number of repetitions), a novelty response could be generated.

Based on our result that inhibitory plasticity is the underlying mechanism of adapted and novelty responses in our model, we wondered how fast it needs to be. Hence, we tested the influence of the inhibitory learning rate ($\eta$) in the unique sequence stimulation paradigm. We found that inhibitory plasticity needs to be fast for both results, the generation of a novelty response (*Figure 4—figure supplement 2A,B*) and adaptation to repeated stimuli (*Figure 4—figure supplement 2C*). Whether such fast inhibitory plasticity operates in the sensory cortex to underlie the adapted and novelty responses is still unknown.

In summary, we identified the plasticity of connections from inhibitory to excitatory neurons belonging to a stimulus-specific assembly as the key mechanism in our framework for the generation of novelty responses and for the resulting adaptation of the network response to repeated stimuli. This adaptation occurs on multiple timescales, covering the range from the timescale of inhibitory plasticity (milliseconds) to sequence block adaptation (seconds) to the presentation of multiple sequence blocks (minutes).

## The adapted response depends on the interval between stimulus presentations

Responses to repeated stimuli do not stay adapted but can recover if the repeated stimulus is no longer presented (*Ulanovsky et al., 2004*; *Cohen-Kashi Malina et al., 2013*). We investigated the

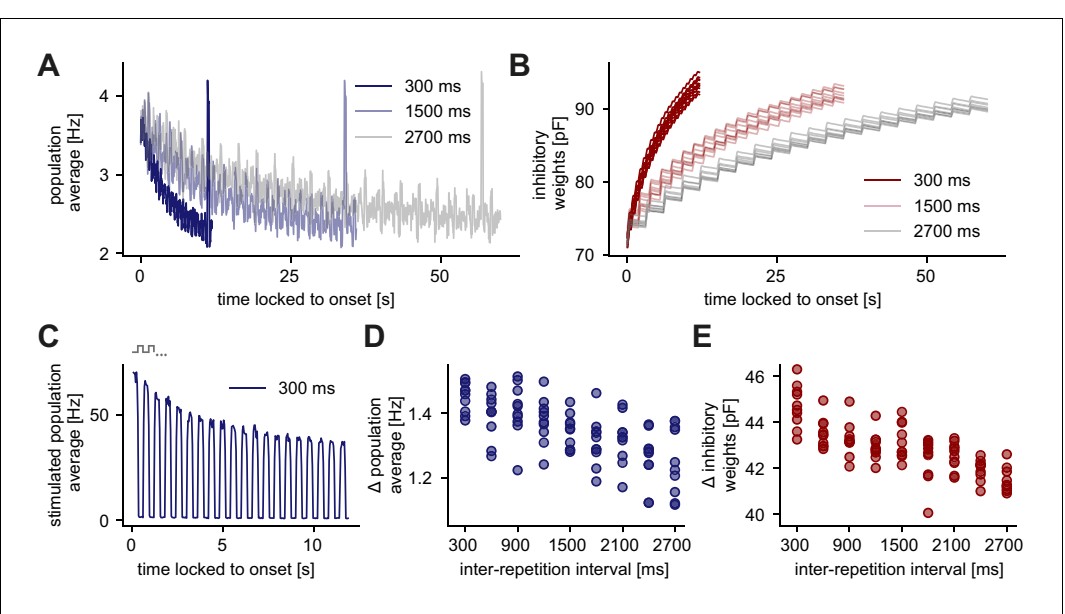

**Figure 5.** Longer inter-repetition intervals decrease the level of adaptation due to the recovery of inhibitory synaptic weights. (**A**) Population average firing rate of all excitatory neurons in the unique sequence stimulation paradigm for varying inter-repetition intervals (varying sequence length). Time is locked to the sequence block onset. (**B**) Evolution of the average inhibitory weights onto stimulus-specific assembly A (identical in all runs) for varying inter-repetition intervals. Time is locked to the sequence block onset. (**C**) Population average firing rate of stimulated excitatory neurons for a 300 ms inter-repetition interval. Time is locked to the sequence block onset. One step in the schematic corresponds to one stimulus in a presented sequence. (**D**) Difference of the onset population rate (measured at the onset of the stimulation, averaged across runs) and the baseline rate (measured before novelty response) as a function of the inter-repetition interval. (**E**) Absolute change of inhibitory weights onto stimulus-specific assembly A from the start until the end of a sequence block presentation as a function of inter-repetition interval.

recovery of adapted responses in the unique sequence stimulation paradigm (*Figure 5A*). Similar to *Figure 2C*, we changed the number of stimuli in the sequence, which leads to different inter-repetition intervals of a repeated sequence stimulus (the interval until the same stimulus is presented again). For example, if two repeated stimuli (A, B) are presented, the inter-repetition interval for each stimulus is 300 ms because each stimulus is presented for 300 ms (*Figure 5C*). If four repeated stimuli are presented (A, B, C, D), the inter-repetition interval for each stimulus is 900 ms. We defined the adaptation level as the difference of the onset population rate, measured at the onset of the stimulation, and the baseline rate, measured shortly before the presentation of a novel stimulus. We found that an increase in the inter-repetition interval reduced the adaptation level of the excitatory population (*Figure 5A,D*) due to a decrease of inhibitory synaptic strength onto stimulus-specific assemblies (*Figure 5B,E*). More specifically, the population average of all excitatory neurons tuned to stimulus A was high when stimulus A was presented and low when stimulus B was presented (*Figure 5C*). Hence, inhibitory weights onto stimulus-specific assembly A increased while A was presented and decreased otherwise (*Figure 5B*).

In summary, longer inter-repetition intervals provide more time for the inhibitory weights onto stimulus-specific assemblies to decrease, hence, weakening the adaptation.

## Inhibitory plasticity and tuned inhibitory neurons support stimulus-specific adaptation

Next, we investigated whether inhibitory plasticity of tuned inhibitory neurons support additional computational capabilities beyond the generation of novelty responses and adaptation of responses to repeated stimuli on multiple timescales. Therefore, we implemented a different stimulation paradigm to investigate the phenomenon of stimulus-specific adaptation (SSA). At the single-cell level, SSA typically involves a so-called oddball paradigm where two stimuli elicit an equally strong response when presented in isolation, but when one is presented more frequently, the elicited response is weaker than for a rarely presented stimulus (*Natan et al., 2015*).

We implemented a similar paradigm at the network level where the excitatory neurons corresponding to two stimuli A and B were completely overlapping and the inhibitory neurons were partially overlapping (*Figure 6A*). Upon presenting stimulus A several times, the neuronal response gradually adapted to the baseline level of activity, while presenting the oddball stimulus B resulted in an increased population response (*Figure 6B*). Therefore, this network was able to generate SSA. Even though stimuli A and B targeted the same excitatory cells, the network response adapted only to stimulus A, while generating a novelty response for stimulus B. Even after presenting stimulus B, activating stimulus A again preserved the adapted response (*Figure 6B*). This form of SSA exhibited by our model network is in agreement with many experimental findings in the primary auditory cortex, primary visual cortex, and multiple other brain areas and animal models (*Nelken, 2014*). In our model network, SSA could neither be generated with adaptive neurons and static synapses (*Figure 6C*, top; Materials and methods), nor with inhibitory plasticity without inhibitory tuning (*Figure 6C*, bottom). In fact, including an adaptive current in the model neurons (*Brette and Gerstner, 2005*) did not even lead to adaptation of the response to a frequent stimulus since firing rates rapidly adapted during stimulus presentation and completely recovered in the inter-stimulus pause (*Figure 6C*, top).

We investigated the dynamics of inhibitory weights to understand the mechanism behind SSA in our model network. During the presentation of stimulus A, stimulus-specific inhibitory weights corresponding to stimulus A (average weights from inhibitory neurons tuned to stimulus A onto excitatory neurons tuned to stimulus A, see *Figure 1—figure supplement 2A*, right) increased their strength, while stimulus-specific inhibitory weights corresponding to stimulus B remained low (*Figure 6D*). Hence, upon presenting the oddball stimulus B, the stimulus-specific inhibitory weights corresponding to stimulus B remained sufficiently weak to keep the firing rate of excitatory neurons high, thus resulting in a novelty response.

We next asked about the recovery of the adapted response in this SSA paradigm (*Figure 6—figure supplement 1*). After a 9 s pause, the response remained adapted (*Figure 6—figure supplement 1A*). Only after more than 200 s the response fully recovered (*Figure 6—figure supplement 1B*). In contrast to the results in *Figure 5*, here, the adaptation level remained high due to the absence of network activity between stimulus presentations. Adaptation slowly recovered as the time between stimulus presentations increased.

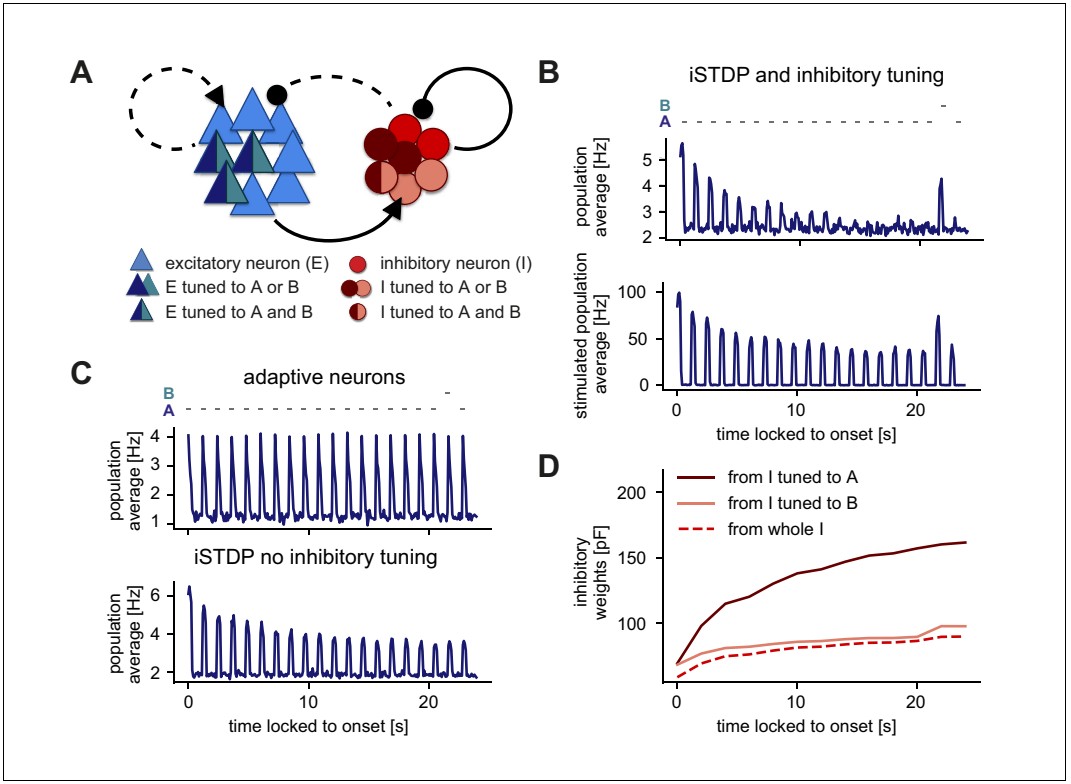

**Figure 6.** Stimulus-specific adaptation follows from inhibitory plasticity and tuning of both excitatory and inhibitory neurons. (**A**) Stimuli A and B provided input to the same excitatory neurons (dark blue and turquoise). Some neurons in the inhibitory population were driven by both A and B (dark red and rose) and some by only one of the two stimuli (dark red or rose). (**B,C**) Population average firing rate of excitatory neurons over time while stimulus A was presented 20 times. Stimulus B was presented instead of A as the second-to-last stimulus. Time is locked to stimulation onset. (**B**) Top: Population average of all excitatory neurons in the network with inhibitory plasticity (iSTDP) and inhibitory tuning. Bottom: Population average of stimulated excitatory neurons only (stimulus-specific to A and B). (**C**) Top: Same as panel B (top) for neurons with an adaptive current in a non-plastic recurrent network. Bottom: Same as panel B (top) for the network with inhibitory plasticity (iSTDP) and no inhibitory tuning. (**D**) Weight evolution of stimulus-specific inhibitory weights corresponding to stimuli A and B and average inhibitory weights.

The online version of this article includes the following figure supplement(s) for figure 6:

**Figure supplement 1.** Recovery of adapted responses in the SSA paradigm.

In summary, our results suggest that the combination of inhibitory plasticity and inhibitory tuning can give rise to SSA. Previous work has argued that inhibition or inhibitory plasticity does not allow for SSA (*Nelken, 2014*). However, this is only true if inhibition is interpreted as a 'blanket' without any tuning in the inhibitory population. Including recent experimental evidence for tuned inhibition into the model, (*Lee et al., 2014*; *Xue et al., 2014*; *Znamenskiy et al., 2018*), can indeed capture the emergence of SSA.

## Disinhibition leads to novelty response amplification and a dense population response

Beyond the bottom-up computations captured by the network response to the different stimuli, we next explored the effect of additional modulations or top-down feedback into our network model. Top-down feedback has been frequently postulated to signal the detection of an error or irregularity in the framework of predictive coding (*Clark, 2013*; *Spratling, 2017*). Therefore, we specifically tested the effect of disinhibitory signals on sequence violations by inhibiting the population of inhibitory neurons during the presentation of a novel stimulus (*Figure 7A*). Recent evidence has identified a differential disinhibitory effect in sensory cortex in the context of adapted and novelty responses

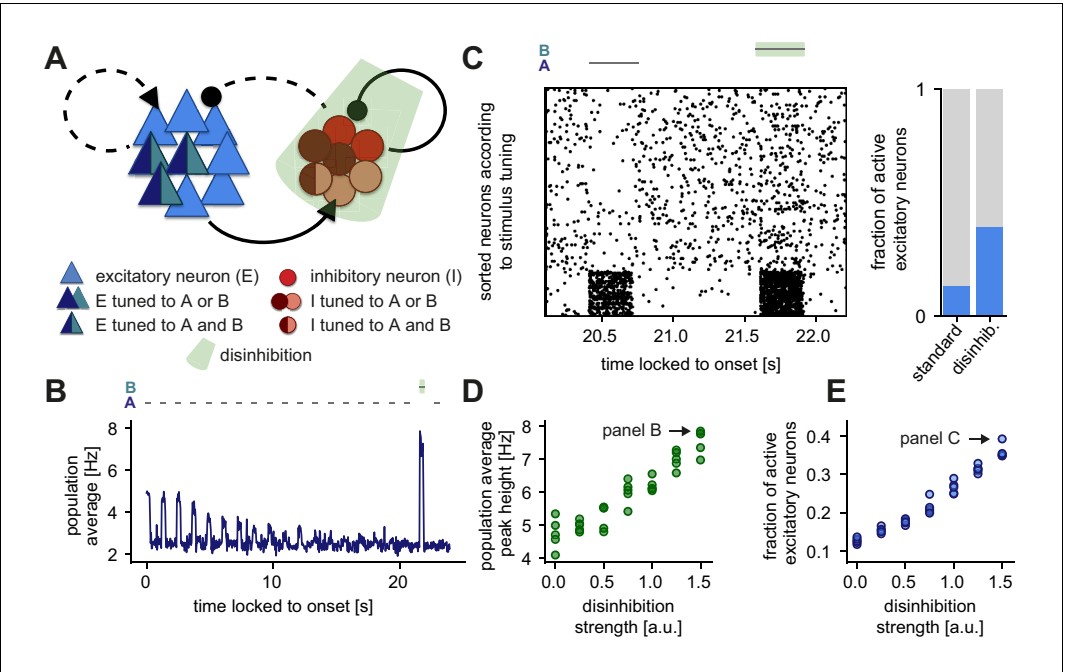

**Figure 7.** Disinhibition leads to a novelty response amplification and a dense population response. (**A**) Stimuli A and B provided input to the same excitatory neurons (dark blue and turquoise). Some neurons in the inhibitory population were driven by both A and B (dark red and rose) and some by only one of the two stimuli (dark red or rose). Inhibition (light green) of the entire inhibitory population led to disinhibition of the excitatory population. (**B**) Population average firing rate of all excitatory neurons over time while stimulus A is presented 20 times. Stimulus B was presented instead of A as the second-to-last stimulus. During the presentation of B, the inhibitory population was inhibited. Time is locked to stimulation onset. (**C**) Left: Raster plot of 250 excitatory neurons corresponding to the population average shown in panel B. The 50 neurons in the bottom part of the raster plot were tuned to stimuli A and B. Time is locked to stimulation onset. Right: Fraction of active excitatory neurons (at least one spike in a 100 ms window) measured directly after the onset of a stimulus. The raster plot and the fraction of active excitatory neurons are shown for the presentation of stimulus B (with disinhibition) and the preceding presentation of stimulus A (standard). (**D**) Population average peak height during disinhibition and the presentation of stimulus B, as a function of the disinhibition strength. Arrow indicates the population average peak height of the trace shown in panel B. Results are shown for five simulations. (**E**) Fraction of active excitatory neurons during disinhibition as a function of the disinhibition strength. Arrow indicates the data point corresponding to panel C. Results are shown for five simulations.

(**Natan et al., 2015**). However, due to the scarcity of detailed knowledge about higher order feed-back signals or within-layer modulations in this context, we did not directly model the source of disinhibition.

When repeating the SSA experiment (**Figure 6**) and applying such a disinhibitory signal (inhibition of the inhibitory population) at the time of the novel stimulus B, our model network amplified the novelty response (**Figure 7B**, shaded green, also compare to **Figure 6B**, top). Disinhibition also increased the density of the network response which corresponds to the number of active excitatory neurons (**Figure 7C**, left). Indeed, disinhibition increased the fraction of active excitatory neurons, which we defined as the fraction of neurons that spike at least once in a 100 ms window during the presentation of a stimulus (**Figure 7C**, right). Dense novelty responses have been recently reported experimentally, where novel stimuli elicited excess activity in a large fraction of the neuronal popula-tion in mouse V1 (**Homann et al., 2017**). Without a disinhibitory signal, the fraction of active neurons for a novel stimulus in our model was qualitatively similar as for repeated stimuli and therefore there was no dense novelty response (**Figure 1—figure supplement 4A**). Given that the inclusion of a dis-inhibitory signal readily increases the density of the novelty response, we suggest that disinhibition might underlie these experimental findings.

In sum, we found that by controlling the total disinhibitory strength (Materials and methods), disinhibition can flexibly amplify the novelty peak (*Figure 7D*) and increase the density of novelty responses (*Figure 7E*). Therefore, we propose that disinhibition can be a powerful mechanism to modulate novelty responses in a network of excitatory and inhibitory neurons.

## Discussion

We developed a recurrent network model with plastic synapses to unravel the mechanistic underpinning of adaptive phenomena and novelty responses. Using the paradigm of repeated stimulus sequences (*Figure 1A*, right), our model network captured the adapted, sparse and periodic responses to repeated stimuli (*Figure 1B–D*) as observed experimentally (*Fairhall, 2014*; *Homann et al., 2017*). The model network also exhibited a transient elevated population response to novel stimuli (*Figure 1B*), which could be modulated by the number of sequence repetitions and the sequence length in the stimulation paradigm (*Figure 2*), in good qualitative agreement with experimental data (*Homann et al., 2017*). We proposed inhibitory synaptic plasticity as a key mechanism behind the generation of these novelty responses. In our model, repeated stimulus presentation triggered inhibitory plasticity onto excitatory neurons selective to the repeated stimulus, reducing the response of excitatory neurons and resulting in their adaptation (*Figure 4*). In contrast, for a novel stimulus inhibitory input onto excitatory neurons tuned to that stimulus remained low, generating the elevated novelty response. Furthermore, we showed that longer inter-repetition intervals led to the recovery of adapted responses (*Figure 5*).

Based on experimental evidence (*Ohki and Reid, 2007*; *Griffen and Maffei, 2014*), we included specific input onto both the excitatory and the inhibitory populations (*Figure 1A*, left). Such tuned inhibition (as opposed to untuned, 'blanket' inhibition commonly used in previous models) enabled the model network to generate SSA (*Figure 6*). Additionally, in the presence of tuned inhibition, a top-down disinhibitory signal achieved a flexible control of the amplitude and density of novelty responses (*Figure 7*). Therefore, besides providing a mechanistic explanation for the generation of adapted and novelty responses to repeated and novel sensory stimuli, respectively, our network model enabled us to formulate multiple experimentally testable predictions, as we describe below.

### Inhibitory plasticity as an adaptive mechanism

We proposed inhibitory plasticity as the key mechanism that allows for adaptation to repeated stimulus presentation and the generation of novelty responses in our model. Many experimental studies have characterized spike-timing-dependent plasticity (STDP) of synapses from inhibitory onto excitatory neurons (*Holmgren and Zilberter, 2001*; *Woodin et al., 2003*; *Haas et al., 2006*; *Maffei et al., 2006*; *Wang and Maffei, 2014*; *D'amour and Froemke, 2015*; *Field et al., 2020*). In theoretical studies, network models usually include inhibitory plasticity to dynamically stabilize recurrent network dynamics (*Vogels et al., 2011*; *Litwin-Kumar and Doiron, 2014*; *Zenke et al., 2015*). In line with recent efforts to uncover additional functional roles of inhibitory plasticity beyond the stabilization of firing rates (*Hennequin et al., 2017*), here, we investigated potential functional consequences of inhibitory plasticity in adaptive phenomena. We were inspired by recent experimental work in the mammalian cortex (*Chen et al., 2015*; *Kato et al., 2015*; *Natan et al., 2015*; *Hamm and Yuste, 2016*; *Natan et al., 2017*; *Heintz et al., 2020*), and simpler systems, such as *Aplysia* (*Fischer et al., 1997*; *Ramaswami, 2014*) and in *Drosophila* (*Das et al., 2011*; *Glanzman, 2011*) along with theoretical reflections (*Ramaswami, 2014*; *Barron et al., 2017*), which all point towards a prominent role of inhibition and inhibitory plasticity in the generation of the MMN, SSA, and habituation. For example, Natan and colleagues observed that in the mouse auditory cortex, both PV and SOM interneurons contribute to SSA (*Natan et al., 2015*), possibly due to inhibitory potentiation (*Natan et al., 2017*). In the context of habituation, daily passive sound exposure has been found to lead to an upregulation of the activity of inhibitory neurons (*Kato et al., 2015*). Furthermore, increased activity to a deviant stimulus in the MMN is diminished when inhibitory neurons are suppressed (*Hamm and Yuste, 2016*).

Most experimental studies on inhibition in adaptive phenomena have not directly implicated inhibitory plasticity as the relevant mechanism. Instead, some studies have suggested that the firing rate of the inhibitory neurons changes, resulting in more inhibitory input onto excitatory cells, effectively leading to adaptation (*Kato et al., 2015*). In principle, there can be many other reasons why

the inhibitory input increases: disinhibitory circuits, modulatory signals driving specific inhibition, or increased synaptic strength of excitatory-to-inhibitory connnections, to name a few. However, following experimental evidence (*Natan et al., 2017*) and supported by our results, the plasticity of inhibitory-to-excitatory connections emerges as a top candidate underlying adaptive phenomena. In our model, adaptation to repeated stimuli and the generation of novelty responses via inhibitory plasticity do not depend on the exact shape of the inhibitory STDP learning rule. It is only important that inhibitory plasticity generates a 'negative feedback' whereby high excitatory firing rates lead to net potentiation of inhibitory synapses while low excitatory firing rates lead to net depression of inhibitory synapses. Other inhibitory STDP learning rules can also implement this type of negative feedback (*Luz and Shamir, 2012*; *Kleberg et al., 2014*), and we suspect that they would also generate the adapted and novelty responses as in our model.

One line of evidence to speak against inhibitory plasticity argues that SSA might be independent of NMDA activation (*Farley et al., 2010*). Inhibitory plasticity, on the contrary, seems to be NMDA receptor-dependent (*D'amour and Froemke, 2015*; *Field et al., 2020*). However, there exists some discrepancy in how exactly NMDA receptors are involved in SSA (*Ross and Hamm, 2020*), since blocking NMDA receptors can disrupt the MMN (*Tikhonravov et al., 2008*; *Chen et al., 2015*). These results indicate that a further careful disentanglement of the underlying cellular mechanisms of adaptive phenomena is needed.

In our model, the direction of inhibitory weight change (iLTD or iLTP) depends on the firing rate of the postsynaptic excitatory cells (see *Vogels et al., 2011*). Postsynaptic firing rates above a 'target firing rate' will on average lead to iLTP, while postsynaptic firing rates below the target rate will lead to iLTD. In turn, the average magnitude of inhibitory weight change depends on the firing rate of the presynaptic inhibitory neurons (see *Vogels et al., 2011*). Therefore, if the background activity between stimulus presentations in our model is very low, recovery from adaptation will only happen on a very slow timescale (as in *Figure 6—figure supplement 1*). However, if the activity between stimulus presentations is higher (either because of a higher background firing rate or because of evoked activity from other sources, for example other stimuli), the adapted stimulus can recover faster (as in *Figure 5*). Therefore, we conclude that our model can capture the reduced adaptation for longer inter-stimulus intervals as found in experiments (*Ulanovsky et al., 2004*; *Cohen-Kashi Malina et al., 2013*) when background activity in the inter-stimulus interval is elevated.

## Alternative mechanisms can account for adapted and novelty responses

Undoubtedly, mechanisms other than inhibitory plasticity might underlie the difference in network response to repeated and novel stimuli. These mechanisms can be roughly summarized in two groups: mechanisms which are unspecific, and mechanisms which are specific to the stimulus. Two examples of unspecific mechanisms are intrinsic plasticity and an adaptive current. Intrinsic plasticity is a form of activity-dependent plasticity, adjusting the neurons' intrinsic excitability (*Debanne et al., 2019*) and has been suggested to explain certain adaptive phenomena (*Levakova et al., 2019*). Other models at the single neuron level incorporate an additional current variable, the adaptive current, which increases for each postsynaptic spike and decreases otherwise. This adaptive current leads to a reduction of the neuron's membrane potential after a spike (*Brette and Gerstner, 2005*). However, any unspecific mechanism can only account for firing-rate adaptation but not for SSA (*Nelken, 2014*; *Figure 6C*). Examples of stimulus-specific mechanisms are short-term plasticity and long-term plasticity of excitatory synapses. Excitatory short-term depression, usually of thalamocortical synapses, is the most widely hypothesized mechanism to underlie adaptive phenomena in cortex (*Nelken, 2014*).

Short-term plasticity (*Abbott, 1997*; *Tsodyks et al., 1998*) has been implicated in a number of adaptation phenomena in different sensory cortices and contexts. One example is an already established model to explain SSA, namely the 'Adaptation of Narrowly Tuned Modules' (ANTM) model (*Nelken, 2014*; *Khouri and Nelken, 2015*). This model has been extensively studied in the context of adaptation to tone frequencies (*Mill et al., 2011a*; *Taaseh et al., 2011*; *Mill et al., 2012*; *Hershenhoren et al., 2014*). Models based on short-term plasticity have also been extended to recurrent networks (*Yarden and Nelken, 2017*) and multiple inhibitory sub-populations (*Park and Geffen, 2020*). Experimental work has shown that short-term plasticity can be different at the synapses from PV and SOM interneurons onto pyramidal neurons, and can generate diverse temporal responses (facilitated, depressed and stable responses) in pyramidal neurons in the auditory cortex

(*Seay et al., 2020*). Short-term plasticity can also capture the differences in responses to periodic versus random presentation of repeated stimuli in a sequence (*Yaron et al., 2012*; *Chait, 2020*). Finally, short-term plasticity has been suggested to explain a prominent phenomenon in the auditory cortex, named 'forward masking' (*Brosch and Schreiner, 1997*), in which a preceding masker stimulus influences the response to a following stimulus (*Phillips et al., 2017*). This highlights short-term plasticity as a key player in adaptive processes in the different sensory cortices, although it likely works in tandem with long-term plasticity.

## Timescales of plasticity mechanisms

The crucial parameter for the generation of adaptation based on short-term plasticity is the timescale of the short-term plasticity mechanism. Experimental studies find adaptation timescales from hundreds of milliseconds to tens of seconds (*Ulanovsky et al., 2004*; *Lundstrom et al., 2010*; *Homann et al., 2017*; *Latimer et al., 2019*), and in the case of habituation even multiple days (*Haak et al., 2014*; *Ramaswami, 2014*). At the same time, the timescales of short-term plasticity can range from milliseconds to minutes (*Zucker and Regehr, 2002*). Hence, explaining the different timescales of adaptive phenomena would likely require a short-term plasticity timescale that can be dynamically adjusted. Our work shows that inhibitory plasticity can readily lead to adaptation on multiple timescales without the need for any additional assumptions (*Figure 4*). However, it is unclear whether inhibitory plasticity can act sufficiently fast to explain adaptation phenomena on the timescale of seconds, as in our model (*Figure 4C,D*). Most computational models of recurrent networks with plastic connections rely on fast inhibitory plasticity to stabilize excitatory rate dynamics (*Sprekeler, 2017*; *Zenke et al., 2017*). Decreasing the learning rate of inhibitory plasticity five-fold eliminates the adaptation to repeated stimuli and the novelty response in our model (*Figure 4—figure supplement 2*). Experimentally, during the induction of inhibitory plasticity, spikes are paired for several minutes and it takes several tens of minutes to reach a new stable baseline of inhibitory synaptic strength (*D'amour and Froemke, 2015*; *Field et al., 2020*). Nonetheless, inhibitory postsynaptic currents increase significantly immediately after the induction of plasticity (see e.g. *D'amour and Froemke, 2015*; *Field et al., 2020*). This suggests that changes of inhibitory synaptic strength already occur while the plasticity induction protocol is still ongoing. Hence, we propose that inhibitory long-term plasticity is a suitable, though not the only, candidate to explain the generation of novelty responses and adaptive phenomena over multiple timescales.

## Robustness of the model

We probed our findings against key parameters and assumptions in our model. First, we tested if the specific choice of pretraining parameters and complexity of presented stimuli affects the generation of adapted and novelty responses. Varying the pretraining duration and the number of pretraining stimuli did not qualitatively change the novelty response and its properties (*Figure 4—figure supplement 1*). In addition, presenting different stimuli in the stimulation phase compared to the pretraining phase (Materials and methods) to mimic the scenario of randomly oriented Gabor patches in *Homann et al., 2017*, preserved the adaptation to repeated stimuli and the generation of a novelty response (*Figure 1—figure supplement 3*).

Second, we explored how the timescale of inhibitory plasticity and of the normalization mechanism affects the generation of adapted and novelty responses. In many computational models, normalization mechanisms are often justified by experimentally observed synaptic scaling. In our model, like in most computational work, the timescale of this normalization was much faster than synaptic scaling (*Zenke et al., 2017*). However, slowing normalization down did not affect the generation of adapted and novelty responses (*Figure 1—figure supplement 5*). Since the change in inhibitory synaptic weights through iSTDP is the key mechanism behind the generation of adapted and novelty responses, the speed of normalization was not crucial as it only affected the excitatory and not the inhibitory weights. In contrast, we found that the learning rate of inhibitory plasticity needs to be 'sufficiently fast' . Slow inhibitory plasticity failed to homeostatically stabilize firing rates in the network. Hence, the network no longer showed an adapted response to repeated stimuli and novelty responses became indiscernible from noise (*Figure 4—figure supplement 2*).

## Disinhibition as a mechanism for novelty response amplification

Upon including a top-down disinhibitory signal in our model network, we observed: (1) an active amplification of the novelty response (*Figure 7B*); (2) a dense novelty response (*Figure 7C*), similar to experimental findings (*Homann et al., 2017*) (without a disinhibitory signal, the novelty response was not dense, see *Figure 1—figure supplement 4*); and (3) a flexible manipulation of neuronal responses through a change in the disinhibitory strength (*Figure 7D,E*).

In our model, we were agnostic to the mechanism that generates disinhibition. However, at least two possibilities exist in which the inhibitory population can be regulated by higher-order feedback to allow for disinhibition. First, inhibitory neurons in primary sensory areas can be shaped by diverse neuromodulatory signals, which allow for subtype-specific targeting of inhibitory neurons (*Froemke, 2015*). Second, higher order feedback onto layer 1 inhibitory cells could mediate the behavioral relevance of the adapted stimuli through a disinhibitory pathway (*Letzkus et al., 2011*; *Wang and Yang, 2018*). Hence, experiments that induce disinhibition either via local mechanisms within the same cortical layer or through higher cortical feedback can provide a test for our postulated role for disinhibition.

In our model, the disinhibitory signal was activated instantaneously. If such additional feedback signals do indeed exist in the brain that signal the detection of higher-order sequence violations, we expect them to arise with a certain delay. Carefully exploring if the dense responses arise with a temporal delay accounting for higher-order processing and projection back to primary sensory areas might shed light on distributed computations upon novel stimuli. These experiments would probably require recording methods on a finer temporal scale than calcium imaging.

Experimental data which points towards a flexible modulation of novelty and adapted responses already exists. The active amplification of novelty responses generated by our model is consistent with some experimental data (*Taaseh et al., 2011*; *Hershenhoren et al., 2014*; *Hamm and Yuste, 2016*; *Harms et al., 2016*), but see also *Vinken et al., 2017*. Giving a behavioral meaning to a sound through fear conditioning has been shown to modify SSA (*Yaron et al., 2020*). Similarly, contrast adaptation has been shown to reverse when visual stimuli become behaviorally relevant (*Keller et al., 2017*). Other studies have also shown that as soon as a stimulus becomes behaviorally relevant, inhibitory neurons decrease their response and therefore disinhibit adapted excitatory neurons (*Kato et al., 2015*; *Makino and Komiyama, 2015*; *Hattori et al., 2017*). Attention might lead to activation of the disinhibitory pathway, allowing for a change in the novelty response compared to the unattended case, as suggested in MMN studies (*Sussman et al., 2014*). Especially in habituation, the idea that a change in context can assign significance to a stimulus and therefore block habituation, leading to 'dehabituation', is widely accepted (*Ramaswami, 2014*; *Barron et al., 2017*).

Hence, we suggest that disinhibition is a flexible mechanism to control several aspects of novelty responses, including the density of the response, which might be computationally important in signaling change detection to downstream areas (*Homann et al., 2017*). Altogether, our results suggest that disinhibition is capable of accounting for various aspects of novelty responses that cannot be accounted for by bottom-up computations. The functional purpose of a dense response to novel stimuli are yet to be explored.

## Functional implications of adapted and novelty responses

In theoretical terms, our model is an attractor network. It differs from classic attractor models where inhibition is considered unspecific (like a 'blanket') (*Amit and Brunel, 1997*). Computational work is starting to uncover the functional role of specific inhibition in static networks (*Rost et al., 2018*; *Najafi et al., 2020*; *Rostami et al., 2020*) as well as the plasticity mechanisms that allow for specific connectivity to emerge (*Mackwood et al., 2021*). These studies have argued that inhibitory assemblies can improve the robustness of attractor dynamics (*Rost et al., 2018*) and keep a local balance of excitation and inhibition (*Rostami et al., 2020*). We showed that specific inhibitory connections readily follow from a tuned inhibitory population (*Figure 1A*, *Figure 1—figure supplement 2*). Our results suggest that adaptation is linked to a stimulus-specific excitatory/inhibitory (E/I) balance. Presenting a novel stimulus leads to a short-term disruption of the E/I balance, triggering inhibitory plasticity, which aims to restore the E/I balance (*Figure 4*; *Vogels et al., 2011*; *D'amour and Froemke, 2015*; *Field et al., 2020*). Disinhibition, which effectively disrupts the E/I balance, allows for flexible control of adapted and novelty responses (*Figure 7*). This links to the notion of

disinhibition as a gating mechanism for learning and plasticity (*Froemke et al., 2007*; *Letzkus et al., 2011*; *Kuhlman et al., 2013*).

A multitude of functional implications have been suggested for the role of adaptation (*Weber et al., 2019*; *Snow et al., 2017*). We showed that one of these roles, the detection of unexpected (or novel) events, follows from the lack of selective adaptation to those events. A second, highly considered functional implication is predictive coding. In the predictive coding framework, the brain is viewed as an inference or a prediction machine. It is thought to generate internal models of the world which are compared to the incoming sensory inputs (*Bastos et al., 2012*; *Clark, 2013*; *Friston, 2018*). According to predictive coding, the overall goal of our brain is to minimize the prediction error, that is the difference between the internal prediction and the sensory input (*Rao and Ballard, 1999*; *Clark, 2013*; *Friston, 2018*). Most predictive coding schemes hypothesize the existence of two populations of neurons. First, prediction error units that signal a mismatch between the internal model prediction and the incoming sensory stimuli. And second, a prediction population unit that reflects what the respective layer 'knows about the world' (*Rao and Ballard, 1999*; *Clark, 2013*; *Spratling, 2017*). Our model suggests that primary sensory areas allow for bottom-up detection of stimulus changes without the need for an explicit population of error neurons or an internal model of the world. However, one could also interpret the state of all inhibitory synaptic weights as an implicit internal model of the recent frequency of various events in the environment.

## Predictions and outlook

Our approach to mechanistically understand the generation of adapted and novelty responses leads to several testable predictions. First, the most general implication from our study is that inhibitory plasticity might serve as an essential mechanism underlying many adaptive phenomena. Our work suggests that inhibitory plasticity allows for adaptation on multiple timescales, ranging from the adaptation to sequence blocks on the timescale of seconds to slower adaptation on the timescale of minutes, corresponding to repeating multiple sequence blocks (*Figure 4C,D*). A second prediction follows from the finding that both excitatory and inhibitory neuron populations show adaptive behavior and novelty responses (*Figure 3B,C*). Adaptation of inhibitory neurons on the single-cell level has already been verified experimentally (*Chen et al., 2015*; *Natan et al., 2015*). Third, we further predict that a violation of the sequence order does not lead to a novelty response. Therefore, the novelty response should not be interpreted as signaling a violation of the exact sequence structure (*Figure 3D,E*). However, previous work has found a reduction in the response to repeated stimuli if the stimuli are presented periodically, rather than randomly, in a sequence (*Yaron et al., 2012*) (but see *Mehra et al., 2021*). Fourth, the height of the novelty peak in the population average depends on the input drive, where decreasing the input strength decreases the novelty response (*Figure 3F*). This could be tested, for example, in the visual system, by presenting visual stimuli with different contrasts.

In our modeling approach, we did not distinguish between different subtypes of inhibitory neurons. This assumption is certainly an oversimplification. The main types of inhibitory neurons, parvalbumin-positive (PV), somatostatin-positive (SOM), and vasoactive intestinal peptide (VIP) expressing neurons, differ in their connectivity and their hypothesized functional roles (*Tremblay et al., 2016*). This is certainly also true for adaptation, and computational studies have already started to tackle this problem (*Park and Geffen, 2020*; *Seay et al., 2020*). Studies of the influence of inhibitory neurons on adaptation have shown that different interneuron types have unique contributions to adaptation (*Kato et al., 2015*; *Natan et al., 2015*; *Hamm and Yuste, 2016*; *Natan et al., 2017*; *Garrett et al., 2020*; *Heintz et al., 2020*). It would be interesting to explore the combination of microcircuit connectivity of excitatory neurons, PVs, SOMs, and VIPs with subtype-specific short-term (*Seay et al., 2020*; *Phillips et al., 2017*) and long-term inhibitory plasticity mechanisms (*Agnes et al., 2020*) on the generation and properties of novelty responses.

In sum, we have proposed a mechanistic model for the emergence of adapted and novelty responses based on inhibitory plasticity, and the regulation of this novelty response by top-down signals. Our findings offer insight into the flexible and adaptive responses of animals in constantly changing environments, and could be further relevant for disorders like schizophrenia where adapted responses are perturbed (*Hamm et al., 2017*).

## Materials and methods

We built a biologically plausible spiking neuronal network model of the mammalian cortex based on recent experimental findings on tuning, connectivity, and synaptic plasticity. The model consists of 4000 excitatory exponential integrate-and-fire (EIF) neurons and 1000 inhibitory leaky integrate-and-fire (LIF) neurons (*Table 1*). Excitatory (E) and inhibitory (I) neurons were randomly recurrently connected (*Table 2*). Excitatory-to-excitatory and inhibitory-to-excitatory connections were plastic (see below). In addition, excitatory-to-excitatory weight dynamics were stabilized by a homeostatic mechanism (*Fiete et al., 2010*), which preserved the total sum of all incoming synaptic weights into an excitatory neuron. All other synapses in the network were fixed. Both excitatory and inhibitory neurons received an excitatory baseline feedforward input in the form of Poisson spikes. Furthermore, different subsets of excitatory and inhibitory neurons received excess input with elevated Poisson rate to model the presentation of stimuli (see below, *Figure 1A*, left; Table 4).

### Dynamics of synaptic conductances and the membrane potential

The membrane dynamics of each excitatory neuron was modeled as an exponential integrate-and-fire (EIF) neuron model (*Fourcaud-Trocmé et al., 2003*):

$$C\frac{d}{dt}V(t) = -g_L(V(t) - V_{\text{rest}}^E) + g_L\Delta_T\exp\left(\frac{V(t) - V_T}{\Delta_T}\right) - g^{EE}(t)(V(t) - V_{\text{rev}}^E) - g^{EI}(t)(V(t) - V_{\text{rev}}^I), \quad (1)$$

where $V(t)$ is the membrane potential of the modeled neuron, $C$ the membrane capacitance, $g_L$ the membrane conductance, and $\Delta_T$ is the slope factor of the exponential rise. The membrane potential was reset to $V_{\text{reset}}$ once the diverging potential reached the threshold peak voltage $V_{\text{peak}}$. Inhibitory neurons were modeled via a leaky-integrate-and-fire neuron model

$$C\frac{d}{dt}V(t) = -g_L(V(t) - V_{\text{rest}}^I) - g^{IE}(t)(V(t) - V_{\text{rev}}^E) - g^{II}(t)(V(t) - V_{\text{rev}}^I). \quad (2)$$

Once the membrane potential reached the threshold voltage $V_{\text{thr}}$, the membrane potential was reset to $V_{\text{reset}}$. The absolute refractory period was modeled by clamping the membrane voltage of a neuron that just spiked to the reset voltage $V_{\text{reset}}$ for the duration $\tau_{\text{abs}}$. In this study, we did not model additional forms of adaptation, such as adaptive currents or spiking threshold $V_T$ adaptation. To avoid extensive parameter tuning, we used previously published parameter values (*Litwin-Kumar and Doiron, 2014*; *Table 1*).

**Table 1.** Parameters for the excitatory (EIF) and inhibitory (LIF) membrane dynamics (*Litwin-Kumar and Doiron, 2014*).

| Symbol | Description | Value |
|---|---|---|
| $N^E$ | Number of E neurons | 4000 |
| $N^I$ | Number of I neurons | 1000 |
| $\tau^E, \tau^I$ | E, I neuron resting membrane time constant | 20 ms |
| $V_{\text{rest}}^E$ | E neuron resting potential | - 70 mV |
| $V_{\text{rest}}^I$ | I neuron resting potential | - 62 mV |
| $\Delta_T$ | Slope factor of exponential | 2 mV |
| $C$ | Membrane capacitance | 300 pF |
| $g_L$ | Membrane conductance | $C/\tau^E$ |
| $V_{\text{rev}}^E$ | E reversal potential | 0 mV |
| $V_{\text{rev}}^I$ | I reversal potential | - 75 mV |
| $V_{\text{thr}}$ | Threshold potential | - 52 mV |
| $V_{\text{peak}}$ | Peak threshold potential | 20 mV |
| $V_{\text{reset}}$ | E, I neuron reset potential | - 60 mV |
| $\tau_{\text{abs}}$ | E, I absolute refractory period | 1 ms |

**Table 2.** Parameters for feedforward and recurrent connections (*Litwin-Kumar and Doiron, 2014*).

| Symbol | Description | Value |
|---|---|---|
| $p$ | Connection probability | 0.2 |
| $\tau_{\text{rise}}^{E}$ | Rise time for E synapses | 1 ms |
| $\tau_{\text{decay}}^{E}$ | Decay time for E synapses | 6 ms |
| $\tau_{\text{rise}}^{I}$ | Rise time for I synapses | 0.5 ms |
| $\tau_{\text{decay}}^{I}$ | Decay time for I synapses | 2 ms |
| $\bar{r}_{\text{ext}}^{EE}$ | Avg. rate of external input to E neurons | 4.5 kHz |
| $\bar{r}_{\text{ext}}^{JE}$ | Avg. rate of external input to I neurons | 2.25 kHz |
| $J_{\min}^{EE}$ | Minimum E to E synaptic weight | 1.78 pF |
| $J_{\max}^{EE}$ | Maximum E to E synaptic weight | 21.4 pF |
| $J_{0}^{EE}$ | Initial E to E synaptic weight | 2.76 pF |
| $J_{\min}^{EI}$ | Minimum I to E synaptic weight | 48.7 pF |
| $J_{\max}^{EI}$ | Maximum I to E synaptic weight | 243 pF |
| $J_{0}^{EI}$ | Initial I to E synaptic weight | 48.7 pF |
| $J^{IE}$ | Synaptic weight from E to I | 1.27 pF |
| $J^{II}$ | Synaptic weight from I to I | 16.2 pF |
| $J^{EEx}$ | Synaptic weight from external input population to E | 1.78 pF |
| $J^{IEx}$ | Synaptic weight from external input population to I | 1.27 pF |

We compared this model to one where we froze plasticity and included adaptive currents $w_{adapt}$ (*Figure 6C*, top). We modeled this by subtracting $w_{adapt}(t)$ on the right hand side of *Equation 1* (*Brette and Gerstner, 2005*). Upon a spike, $w_{adapt}(t)$ increased by $b_w$ and the sub-threshold dynamics of the adaptive current were described by $\tau_w \frac{d}{dt} w_{\text{adapt}}(t) = -w_{\text{adapt}}(t) + a_w(V(t) - V_{\text{rest}}^{E})$, where $a_w = 4$ nS denotes the subthreshold and $b_w = 80.5$ pA the spike-triggered adaptation. The adaptation time scale was set to $\tau_w = 150$ ms.

The conductance of neuron $i$ which is part of population $X$ and is targeted by another neuron in population $Y$ was denoted with $g_i^{XY}$. Both $X$ and $Y$ could refer either to the excitatory or inhibitory population, that is $X, Y \in [E, I]$. The shape of the synaptic kernels $F(t)$ was a difference of exponentials and differed for excitatory and inhibitory input depending on the rise and decay times $\tau_{\text{decay}}^{Y}$ and $\tau_{\text{rise}}^{Y}$:

$$F^{Y}(t) = \frac{e^{\frac{-t}{\tau_{\text{decay}}^{Y}}} - e^{\frac{-t}{\tau_{\text{rise}}^{Y}}}}{\tau_{\text{decay}}^{Y} - \tau_{\text{rise}}^{Y}}. \tag{3}$$

This kernel was convolved with the total inputs to neuron $i$ weighted with the respective synaptic strength to yield the total conductance

$$g_i^{XY}(t) = F^{Y}(t) * \left( J_{ext}^{XY} s_{i,ext}^{XY}(t) + \sum_j J_{ij}^{XY} s_j^{Y}(t) \right), \tag{4}$$

where $s_j^{Y}(t)$ is the spike train of neuron $j$ in the network and $s_{i,ext}^{XY}$ denotes the spike train of the external input to neuron $i$. The external spike trains were generated in an independent homogeneous Poisson process. The synaptic strength from the input neurons to the network neurons, $J_{ext}^{XY}$, was assumed to be constant.

## Excitatory and inhibitory plasticity

We implemented the plasticity from an excitatory to an excitatory neuron $J^{EE}$ based on the triplet spike-time-dependent plasticity rule (triplet STDP), which uses triplets of pre- and postsynaptic

spikes to evoke synaptic change (*Sjöström et al., 2001*; *Pfister and Gerstner, 2006*). The addition of a third spike for the induction of synaptic plasticity modifies the amount of potentiation and depression induced by the classical pair-based STDP, where pairs of pre- and postsynaptic spikes induce plasticity based on their timing and order (*Bi and Poo, 1998*). The triplet eSTDP rule has been shown to capture the dependency of plasticity on firing rates found experimentally, whereby a high frequency of pre- and postsynaptic spike pairs leads to potentiation rather than no synaptic change as predicted by pair-based STDP (*Sjöström et al., 2001*; *Pfister and Gerstner, 2006*; *Gjorgjieva et al., 2011*; *Table 3*). In the triplet rule, four spike accumulators, $r_1, r_2, o_1,$ and $o_2$, increase by one, once a spike of the corresponding neuron occurs and otherwise decrease exponentially depending on their respective time constant $\tau_+, \tau_x, \tau_-,$ and $\tau_y$:

$$
\begin{aligned}
\frac{dr_1(t)}{dt} &= -\frac{r_1(t)}{\tau_+} \text{ if } t = t^{\mathrm{pre}} \text{ then } r_1 \rightarrow r_1 + 1, \\
\frac{dr_2(t)}{dt} &= -\frac{r_2(t)}{\tau_x} \text{ if } t = t^{\mathrm{pre}} \text{ then } r_2 \rightarrow r_2 + 1, \\
\frac{do_1(t)}{dt} &= -\frac{o_1(t)}{\tau_-} \text{ if } t = t^{\mathrm{post}} \text{ then } o_1 \rightarrow o_1 + 1, \\
\frac{do_2(t)}{dt} &= -\frac{o_2(t)}{\tau_y} \text{ if } t = t^{\mathrm{post}} \text{ then } o_2 \rightarrow o_2 + 1.
\end{aligned}
\tag{5}
$$

The E-to-E weights were updated as

$$
\begin{aligned}
\Delta J^{EE}(t) &= -o_1(t)[A_2^- + A_3^- r_2(t - \epsilon)] \text{ if } t = t^{\mathrm{pre}}, \\
\Delta J^{EE}(t) &= r_1(t)[A_2^+ + A_3^+ o_2(t - \epsilon)] \text{ if } t = t^{\mathrm{post}},
\end{aligned}
\tag{6}
$$

where the $A^+, A^-$ corresponds to the excitatory LTP or LTD amplitude, and the subscript refers to the triplet (3) or pairwise term (2). The parameter $\epsilon > 0$ ensures that the weights are updated prior to increasing the respective spike accumulators by 1. Spike detection was modeled in an all-to-all approach.

The plasticity of inhibitory-to-excitatory connections, $J^{EI}$, was modeled based on a symmetric inhibitory pairwise STDP (iSTDP) rule, initially suggested on theoretical grounds for its ability to homeostatically stabilize firing rates in recurrent networks (*Vogels et al., 2011*). According to this rule, the timing but not the order of pre- and postsynaptic spikes matters for the induction of synaptic plasticity. Other inhibitory rules have also been measured experimentally, including classical Hebbian and anti-Hebbian (e.g. *Holmgren and Zilberter, 2001*; *Woodin et al., 2003*; *Haas et al., 2006*; for a review see *Hennequin et al., 2017*), and some may even depend on the type of the interneuron (*Udakis et al., 2020*). We chose the iSTDP rule because it can stabilize excitatory firing rate dynamics in recurrent networks (*Vogels et al., 2011*; *Litwin-Kumar and Doiron, 2014*) and was

**Table 3.** Parameters for the implementation of Hebbian and homeostatic plasticity (*Pfister and Gerstner, 2006*; *Litwin-Kumar and Doiron, 2014*).

| Symbol | Description | Value |
|---|---|---|
| $\tau_-$ | Time constant of pairwise pre-synaptic detector (+) | 33.7 ms |
| $\tau_+$ | Time constant of pairwise post-synaptic detector (-) | 16.8 ms |
| $\tau_x$ | Time constant of triplet pre-synaptic detector (-) | 101 ms |
| $\tau_y$ | Time constant of triplet post-synaptic detector (+) | 125 ms |
| $A_2^+$ | Pairwise potentiation amplitude | $7.5 \times 10^{-10}$ pF |
| $A_3^+$ | Triplet potentiation amplitude | $9.3 \times 10^{-3}$ pF |
| $A_2^-$ | Pairwise depression amplitude | $7 \times 10^{-3}$ pF |
| $A_3^-$ | Triplet depression amplitude | $2.3 \times 10^{-4}$ pF |
| $\tau_y^{\mathrm{inhib}}$ | Time constant of low-pass filtered spike train | 20 ms |
| $\eta$ | Inhibitory plasticity learning rate | 1 pF |
| $r_0$ | Target firing rate | 3 Hz |

recently verified to operate in the auditory cortex of mice (*D'amour and Froemke, 2015*). The plasticity parameters are shown in *Table 3*. The two spike accumulators $y^{E/I}$, for the inhibitory pre- and the excitatory post-synaptic neuron, have the same time constant $\tau_y^{\text{inhib}}$. Their dynamics were described by

$$
\begin{aligned}
\frac{dy^I(t)}{dt} &= -\frac{y^I(t)}{\tau_y^{\text{inhib}}} \text{ if } t = t^{\text{pre/I}} \text{ then } y^I \to y^I + 1 \text{ and} \\
\frac{dy^E(t)}{dt} &= -\frac{y^E(t)}{\tau_y^{\text{inhib}}} \text{ if } t = t^{\text{post/E}} \text{ then } y^E \to y^E + 1.
\end{aligned}
\tag{7}
$$

The I-to-E weights were updated as

$$
\begin{aligned}
\Delta J_{ij}^{EI}(t) &= \eta(y_i^E(t) - 2r_0\tau_y^{\text{inhib}}) \text{ if } t = t^{\text{pre/I}} \\
\Delta J_{ij}^{EI}(t) &= \eta y_j^I(t) \text{ if } t = t^{\text{post/E}},
\end{aligned}
\tag{8}
$$

where $\eta$ is the learning rate, and $r_0$ corresponds to the target firing rate of the excitatory neuron. In *Figure 4—figure supplement 2* we investigated the inhibitory learning rate $\eta$. *Figure 1—figure supplement 1* shows the excitatory and inhibitory STDP rules for different pairing frequencies.

## Additional homeostatic mechanisms

Inhibitory plasticity alone is considered insufficient to prevent runaway activity in this network implementation. Hence, additional mechanisms were implemented that also have a homeostatic effect. To avoid unlimited weight increase, the synaptic weights were bound from below and from above, see *Table 2*. Subtractive normalization ensured that the total synaptic input to an excitatory neuron remains constant throughout the simulation. This was implemented by scaling all incoming weights to each neuron every $\Delta t = 20$ ms according to

$$
\Delta J_{ij}^{EE}(t) = -\frac{\sum_j J_{ij}^{EE}(t) - \sum_j J_{ij}^{EE}(0)}{N_i^E},
\tag{9}
$$

where $i$ is the index of the post-synaptic and $j$ of the pre-synaptic neurons. $N_i^E$ is the number of excitatory connections onto neuron $i$ (*Fiete et al., 2010*). In *Figure 1—figure supplement 5* we investigated the effect of the normalization timestep $\Delta t$ on the novelty response.

## Stimulation protocol

All neurons received external excitatory baseline input. The baseline input to excitatory neurons $r_{\text{ext}}^E$ was higher than the input to inhibitory neurons $r_{\text{ext}}^I$ (*Table 4*). An external input of $r_{\text{ext}}^E = 4.5$ kHz can be interpreted as 1000 external presynaptic neurons with average firing rates of 4.5 Hz (compare *Litwin-Kumar and Doiron, 2014*).

The stimulation paradigm was inspired by a recent study in the visual system (*Homann et al., 2017*). In *Homann et al., 2017*, the stimulation consisted of images with 100 randomly chosen, superimposed Gabor patches. Rather than explicitly modeling oriented and spatially localized Gabor patches, in our model, stimuli that correspond to Gabor patches of a given orientation were

**Table 4.** Parameters for the stimulation paradigm and stimulus tuning.

| Symbol | Description | Value |
|---|---|---|
| $r_{\text{ext}}^E$ | External baseline input to E | 4.5 kHz |
| $r_{\text{ext}}^I$ | External baseline input to I | 2.25 kHz |
| $r_{\text{stim}}^E$ | Additional input to E during stimulus presentation | 12 kHz |
| $r_{\text{stim}}^I$ | Additional input to I during stimulus presentation | 1.2 kHz |
| $r_{\text{disinh}}^I$ | Additional input to I during disinhibition | −1.5 kHz |
| $p_{\text{member}}^E$ | Probability for an E neuron to be driven by a stimulus | 5% |
| $p_{\text{member}}^I$ | Probability for an I neuron to be driven by a stimulus | 15% |

implemented by simultaneously co-activating subsets of cells by strongly driving them. Hence, the model analog of the presentation of a sensory stimulus, in our experiments, is increased input to a subset of neurons. Every time a particular stimulus is presented again, the same set of neurons receives strong external stimulation, $r_{\text{stim}}^E$ and $r_{\text{stim}}^I$. Therefore, while a stimulus in our stimulation paradigm is functionally similar to presenting Gabor patches with similar orientations, it does not represent the Gabor patches themselves.

We first implemented a pretraining phase. In this phase, we sequentially stimulated subsets of neurons that are driven by all stimuli (repeated and novel stimuli) eventually used in the stimulation phase. The stimuli were presented in random order, leading to a change in network connectivity that is only stimulus but not sequence-dependent (*Figure 4B*, first 100 s shown here for five repetitions of each stimulus). Hence, the pretraining phase is a phenomenological model of the development process to generate a structure in the network connections prior to the actual stimulation paradigm. This can be interpreted as imprinting a 'backbone' of orientation selective neurons, where cells which are selective to similar features (e.g. similar orientations) become strongly connected due to synaptic plasticity (as seen in experiments, see for e.g. *Ko et al., 2011*; *Ko et al., 2013*).

Next, we implemented a stimulation phase where we presented the same stimuli used during the pretraining phase according to the repeated sequence stimulation paradigm. To match the randomly oriented Gabor patches presented in *Homann et al., 2017*, we also performed additional simulations where in the stimulation phase we activated different, randomly chosen, subsets of neurons (*Figure 1—figure supplement 3*) (note that there is some overlap with the imprinted orientation selective subsets).

In the standard repeated sequence stimulation paradigm (*Figure 3* and *Figure 4*), a total of 65 stimuli were presented (5 x 3 repeated + 5 x 10 novel stimuli) during pretraining. In *Figure 4—figure supplement 1*, we tested if changes in the pretraining phase, such as a change in the number of repetitions of each stimulus or the total number of stimuli, affect our results.

The timescales of the experimental paradigm in *Homann et al., 2017* and the model paradigm were matched, that is the neurons tuned to a stimulus received additional input for 300 ms simulation time. Stimuli were presented without pauses in between, corresponding to continuous stimulus presentation without blank images (visual) or silence (auditory) between sequence blocks. *Table 4* lists the stimulus parameters.

In contrast to several previous plastic recurrent networks, we did not only consider the excitatory neurons to have stimulus tuning properties but included inhibitory tuning as well. The probability of an excitatory neuron to be driven by one particular stimulus was 5%, leading to roughly 200 neurons that responded specifically to this stimulus. We modeled inhibitory tuning to be both weaker and broader. The probability of an inhibitory neuron to be driven by one particular stimulus was 15%, leading to roughly 150 neurons that responded specifically to this stimulus. There was overlap in stimulus tuning, that is, one neuron could be driven by multiple stimuli. Given this broader tuning of inhibitory neurons compared to excitatory neurons, a single inhibitory neuron could strongly inhibit multiple excitatory neurons which were selective to different stimuli, effectively implementing lateral inhibition.

Stimulus tuning in both populations led to the formation of stimulus-specific excitatory assemblies due to synaptic plasticity, where the subsets of excitatory neurons receiving the same input developed strong connections among each other as noted above (*Figure 1—figure supplement 2C*) and found experimentally (*Ko et al., 2011*; *Miller et al., 2014*; *Lee et al., 2016*). The strong, bidirectional connectivity among similarly selective neurons in our model was a direct consequence of the triplet STDP rule (*Gjorgjieva et al., 2011*; *Montangie et al., 2020*). Additionally, the connections from similarly tuned inhibitory to excitatory neurons also became stronger, as seen in experiments (*Lee et al., 2014*; *Xue et al., 2014*; *Znamenskiy et al., 2018*; *Najafi et al., 2020*). The number of stimulus-specific assemblies varied depending on the stimulation paradigm and corresponded to the number of unique stimuli presented in a given paradigm. We did not impose topographic organization of these assemblies (for e.g. tonotopy in the auditory cortex) since it would not influence the generation of adapted and novelty responses, but increase model complexity. Such spatial organization could, however, be introduced by allowing the assemblies for neighboring stimuli to overlap.

Disinhibition in the model was implemented via additional inhibiting input to the inhibitory population $r_{\text{inhib}}^I$. This was modeled in a purely phenomenological way, and we are agnostic as to what causes the additional inhibition.

## Simulation details

The simulations were performed using the Julia programming language. Further evaluation and plotting was done in Python. Euler integration was implemented using a time step of 0.1 ms. Code implementing our model and generating the stimulation protocols can be found here: https://github.com/comp-neural-circuits/novelty-via-inhibitory-plasticity (*Schulz, 2021*; copy archived at swh:1:rev:d368b14a2368925b290923c2c11411d7b7a40bd1).

## Acknowledgements

AS, CM, and JG thank the Max Planck Society for funding and MJB thanks the NEI and the Princeton Accelerator Fund for funding. We thank members of the 'Computation in Neural Circuits' group for useful discussions and comments on the manuscript.

## Additional information

### Funding

| Funder | Grant reference number | Author |
| --- | --- | --- |
| Max-Planck-Gesellschaft | Research Group Award to JG | Auguste Schulz Christoph Miehl Julijana Gjorgjieva |
| NEI and Princeton Accelerator Fund | | Michael J Berry |

The funders had no role in study design, data collection and interpretation, or the decision to submit the work for publication.

### Author contributions

Auguste Schulz, Christoph Miehl, Conceptualization, Resources, Software, Formal analysis, Investigation, Visualization, Methodology, Writing - original draft, Writing - review and editing; Michael J Berry II, Conceptualization, Methodology, Writing - review and editing; Julijana Gjorgjieva, Conceptualization, Resources, Supervision, Funding acquisition, Methodology, Writing - original draft, Writing - review and editing

### Author ORCIDs

Auguste Schulz ⓘ https://orcid.org/0000-0001-8616-3756
Christoph Miehl ⓘ http://orcid.org/0000-0001-9094-2760
Michael J Berry II ⓘ https://orcid.org/0000-0003-4133-7999
Julijana Gjorgjieva ⓘ https://orcid.org/0000-0001-7118-4079

### Decision letter and Author response

Decision letter https://doi.org/10.7554/eLife.65309.sa1
Author response https://doi.org/10.7554/eLife.65309.sa2

## Additional files

### Supplementary files

• Transparent reporting form

### Data availability

The code to reproduce the figures for this paper has been uploaded on GitHub and be accessed here: https://github.com/comp-neural-circuits/novelty-via-inhibitory-plasticity (copy archived at https://archive.softwareheritage.org/swh:1:rev:d368b14a2368925b290923c2c11411d7b7a40bd1).

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
