## [Decision Letter]

**Acceptance summary:**

Your paper identifies an important mechanism for generation of novelty signals. It expands current understanding of the functional role of inhibitory plasticity, and makes predictions that can be tested experimentally in future studies.

**Decision letter after peer review:**

Thank you for submitting your article "The generation of cortical novelty responses through inhibitory plasticity" for consideration by *eLife*. Your article has been reviewed by 3 peer reviewers, including Maria N Geffen as Reviewing Editor and Reviewer #1, and the evaluation has been overseen by Joshua Gold as the Senior Editor.

Essential revisions:

1) Please revise the paper to address the specific points raised by all reviewers.

2) Addressing points 1, 3, 4, 5 and 7 by reviewer 1 and points 1, 2, 3 and 5 by reviewer 3 will likely require additional evidence and discussion.

*Reviewer #1 (Recommendations for the authors):*

This is an interesting and well-written paper which presents the results of a model for cortical plasticity and resulting increase in neuronal responses to unexpected stimuli. Overall, this is an elegant study that provides a number of interesting, experimentally testable, hypotheses and develops a prediction for a mechanism for novelty response generation. The results are clearly presented. We have several suggestions that should allow to better integrate the study with known experimental results.

1. Whereas the authors use the term "novelty" response, it is unclear whether this is a true novelty response because a number of stimulus parameters differs from those identified in the literature. The network is not sensitive to temporal structure, which suggests that it does not completely replicate certain aspects of neuronal adaptation in cortex. MEG studies in humans (work from Maria Chait's lab), and Yaron et al., 2012 (SSA under different contexts in rats) all suggest that cortex should exhibit a differential response to different sequences of stimuli drawn from the same distribution. What modifications to the model would produce adaptation to temporal structure of the stimuli?

2. Figure 2 provides support for a key result of Homann et al., 2017. It would be interesting to see if the network behaves similarly with more complex stimuli and generates novelty responses to complex stimuli. In Homann et al., 2017 experiment, the stimuli were scenes comprised of many Gabor patches. These could potentially broadly activate visual cortex as opposed to the "simple" single stimuli used here. It would be interesting to consider how a broader pattern of stimulation would affect the results.

3. In the experimental results, a large majority of individual neurons exhibited the novelty response. Does this also occur in the model? Would it be possible to present statistics and single-neuronal data in addition to the population mean responses?

4. It is unclear how duration of pre-training relates to plasticity and novelty responses (Figure 4B). Would you expect increased pretraining duration affect the novelty response behavior? If so, perhaps the duration of pre-training should be varied systematically? Furthermore, how does the number of pre-training stimuli affect the performance?

5. It is difficult to interpret the differences in adaptation between different repeated stimuli (Figure 4C bottom -> green vs blue lines) as compared to the differences between the "late" and "intermediate" time points (4D bottom). Results depicted in figure 4D suggest multiple timescales of adaptation. Are the differences in 4C, which seem to be of similar magnitude, meaningful?

6. One of the primary findings of Natan et al., 2015 was the differential effect of optogenetic disinhibition of SOM vs PV interneurons on the SSA response. The differential effect is seen when disinhibiting the standards -1, +1, +2 etc. relative to the deviant. It would be interesting to see what the effect of disinhibition is on the network response (Figure 6), and how it relates to Natan's findings.

7. The motivation for having different STDP learning rules for E->E and I->E connections is unclear, other than citation of previous work. Can you please explore in more detail the differences between these learning rules?

8. An emerging theme from this paper and from other models such as Yarden and Nelken, 2017 and Park and Geffen, 2020, may be that the tonotopic organization of similarly tuned neurons helps facilitate adaptation. Tuned assemblies were a key feature of the models in these papers. Here, the tonotopic organization arises from the STDP rule here, and it would be interesting to discuss the relationship between tonotopic organization and plasticity.

9. Seay et al., 2020, have considered plasticity rules in adaptation as a function of facilitation of specific inhibitory interneurons in SSA. It would be interesting to speculate whether and how this model and the learning rules relates to those published results.

*Reviewer #2 (Recommendations for the authors):*

The units of the learning rate n of inhibitory plasticity on page 18 is listed in uF and only very briefly discussed? Since eta seems to be relatively important for the novelty detection, and determines e.g. how many trials a system will typically require to learn a stimulus (see also Figure 2) it should thus be discussed in more detail, possibly with an additional set of simulations that are complementary to those in Figure 2.

The inhibitory learning rule used here is stated to be confirmed on p. 15. This is arguable, and the jury may still be out. It may be worth discussing how other rules would perform?

*Reviewer #3 (Recommendations for the authors):*

1. The abstract is a bit cryptic: it states that "inhibitory synaptic plasticity readily generates novelty responses". Inhibitory plasticity is rather vague, it is necessary to read quite far into the results to figure out if the key is short-term synaptic inhibitory plasticity as in a number of models, or some form of long-term associative synaptic plasticity. The abstract should be more informative and explicitly state that the model relies on STDP of Inh->Ex synapse.

2. It would be helpful to provide a bit more information about the model architecture in Figure 1 and the start of the Results section so the reader does not immediately have to go to the Methods section (e.g., network size, ratio of E/I neurons, no STP).

3. In the Methods it would also be helpful to plot the STDP function for excitatory and inhibitory plasticity for spike pairs.

---

## [Author Response]

Reviewer #1 (Recommendations for the authors):This is an interesting and well-written paper which presents the results of a model for cortical plasticity and resulting increase in neuronal responses to unexpected stimuli. Overall, this is an elegant study that provides a number of interesting, experimentally testable, hypotheses and develops a prediction for a mechanism for novelty response generation. The results are clearly presented. We have several suggestions that should allow to better integrate the study with known experimental results.

We thank the reviewer for her detailed and helpful feedback. We address the raised points in detail below. In short, we added four new supplementary figures related to the reviewer's comments (Figure 1—figure supplement 1,3,4 and Figure 4—figure supplement 1) and substantially rewrote multiple parts of our manuscript.

1. Whereas the authors use the term "novelty" response, it is unclear whether this is a true novelty response because a number of stimulus parameters differs from those identified in the literature. The network is not sensitive to temporal structure, which suggests that it does not completely replicate certain aspects of neuronal adaptation in cortex. MEG studies in humans (work from Maria Chait's lab), and Yaron et al., 2012 (SSA under different contexts in rats) all suggest that cortex should exhibit a differential response to different sequences of stimuli drawn from the same distribution. What modifications to the model would produce adaptation to temporal structure of the stimuli?

To produce adaptation to the exact temporal structure of stimuli in a sequence, our model will most likely need the addition of short-term plasticity.

As the reviewer points out, in some studies [Yaron et al., 2012, Chait, 2020] a novelty response is defined to be sensitive to the stimulus temporal structure. In our work, we use the term `temporal structure' to refer to the periodic structure of the sequence (i.e. a periodic sequence, e.g. ABCABCABC, vs. a non-periodic sequence, e.g. ACBBACBCA). The term `temporal structure' can also include the duration of a stimulus and the interval between stimuli (see also our answer to comment 2 by reviewer 3). To distinguish between these different possibilities, we now use `sequence structure' instead of `temporal structure' to refer to the periodic structure of the sequence. In Figure 3E, we show that the generation of a novelty response does not depend on having a periodic structure of the sequence. A novelty response still occurs when the stimuli in a sequence are shuffled (compare Figure 3E with Figure 3A). This is similar to the findings of [Yaron et al., 2012], where following the presentation of repeated stimuli in a sequence, an elevated response to a novel stimulus emerges independent of whether the repeated stimuli are presented periodically or randomly (although it is somewhat higher in the case of random stimulus presentation). Therefore, we conclude that a periodic sequence is not necessary for the occurrence of a novelty response.

Interestingly, [Yaron et al., 2012] also found a small, but statistically significant, reduction in the adapted response to repeated stimuli when the stimuli were presented periodically rather than randomly in a sequence. This suggests that there might be an additional adaptation to the presentation of periodic stimuli because these stimuli are better predictable. In contrast, a recent preprint found the opposite: higher responses to repeated stimuli when they were presented periodically rather than randomly [Mehra et al., 2021]. Our current model cannot capture the small differences found in [Yaron et al., 2012] because the responses to repeated and novel stimuli depend only on the statistics of the presented stimuli, and not on the exact temporal structure (periodic vs. random). One plausible way to capture the difference between periodic and random stimulus presentation might be to add short-term plasticity to our model. Several computational studies have already shown that short-term plasticity can generate history dependence in the response to a stimulus (see e.g. [Seay et al., 2020, Phillips et al., 2017]). Periodic stimulus presentation could lead to more short-term depression compared to a random presentation of repeated stimuli and therefore lead to a small reduction of average responses. However, combining short-term plasticity and long-term plasticity in recurrent circuits of excitatory and inhibitory neurons is currently beyond the scope of our work. MEG signals recorded in humans, in the context of mismatch-negativity (MMN) are also sensitive to the temporal structure in sound sequences, but to explain these findings more complex models on different scales are needed (see e.g. [Chait, 2020]).

We now define the term `novelty response' in line 44. In lines 502 and 640 in the Discussion we discuss the difference in responses to periodically vs. randomly presented stimuli in a sequence. In line 654 we acknowledge that further studies including both short- and long-term plasticity are needed to fully uncover all aspects of adapted and novelty responses. We also changed the section title to "Stimulus periodicity in the sequence is not required for the generation of a novelty response" and clarified the notion of `sequence structure' throughout the section.

2. Figure 2 provides support for a key result of Homann et al., 2017. It would be interesting to see if the network behaves similarly with more complex stimuli and generates novelty responses to complex stimuli. In Homann et al., 2017 experiment, the stimuli were scenes comprised of many Gabor patches. These could potentially broadly activate visual cortex as opposed to the "simple" single stimuli used here. It would be interesting to consider how a broader pattern of stimulation would affect the results.

More complex stimuli also generate adaptation to repeated stimuli and produce a novelty response to a novel stimulus (Figure 1—figure supplement 3), similar to the `simple' stimuli used in Figure 1-4.

First, we note that in our model, rather than explicitly modeling oriented and spatially-localized Gabor patches, stimuli that correspond to Gabors of a given orientation simultaneously co-activate subsets of neurons by strongly driving them. Hence, the model analog of the presentation of a sensory stimulus in the experiments is increased input to a subset of neurons. Presenting these stimuli in the pretraining phase leads to the imprinting of structure in the recurrent network where neurons which are selective to similar features (e.g. a similar orientation) become strongly connected due to synaptic plasticity (as seen in experiments, see for e.g. [Ko et al., 2011, Ko et al., 2013]). This can be interpreted as imprinting a `backbone' of orientation selective neurons to simulate development, with strong recurrent connections among neurons that share selectivity. Therefore, a stimulus in our pretraining stimulation paradigm is functionally similar to presenting Gabor patches with similar orientations, but does not represent the Gabor patches themselves.

Next, to answer the reviewer's comment, we modified our model to test how it responds to more complex stimuli that correspond to the experiment in [Homann et al., 2017]. As before, we initially simulated development by presenting a set of stimuli during the pretraining phase to generate the `backbone' of orientation selective neurons (Figure 1—figure supplement 3A). In the [Homann et al., 2017] experiment, the visual stimuli consist of 100 randomly oriented Gabor patches. To match these randomly oriented Gabor patches, in the stimulation phase we activated different, randomly chosen, subsets of neurons (note that there is some overlap with the developmentally imprinted subsets). The weight matrix before the stimulation phase does not show any assemblies specific to repeated or novelty stimuli, compared to the weight matrix after the stimulation phase (Figure 1—figure supplement 3E,F, compare left corresponding to before the stimulation phase and right corresponding to after the stimulation phase). We confirmed that these more complex stimuli generate similar results as the `simpler' stimuli used in Figure 1-4. In particular, the more complex stimuli also generate adaptation to repeated stimuli and produce a novelty response to a novel stimulus (Figure 1—figure supplement 3B). As before, the observed results come from an increase of the inhibitory synaptic weights (Figure 1-Figure Supplement 3D, bottom row). In general, the evolution of synaptic weights does not differ qualitatively after the pretraining phase (Figure 1—figure supplement 3C,D, compare with Figure 4B-D).

We present results with these more complex stimuli in a new figure (Figure 1—figure supplement 3) and discuss them in our Results section in lines 151 and 277, the Methods section in line 757 and 776 and the new Discussion section `Robustness of the model' in line 533.

3. In the experimental results, a large majority of individual neurons exhibited the novelty response. Does this also occur in the model? Would it be possible to present statistics and single-neuronal data in addition to the population mean responses?

In the model, the fraction of active excitatory neurons is qualitatively similar for novel and for the adapted and onset stimuli (Figure 1—figure supplement 4), in contrast to the findings of [Homann et al., 2017] (see Justification below). However, the fraction of active neurons increases when a disinhibitory signal is applied (Figure 7C, E).

Homann et al., 2017 found that a repeated stimulus sparsely activates neurons in V1, referred to as a low density or a sparse response, whereas a novel stimulus evokes excess activity in a much larger fraction of the neurons in V1, referred to as a high density or a dense response [Homann et al., 2017]. To quantify the density of the adapted and novel responses in the model, we used the fraction of active neurons as a measure of the density of a response. We call active neurons those that spike at least once within a 100 ms window directly after the onset of the stimulus (onset), after the onset of the novel stimulus (novelty) and shortly before the novel stimulus is presented (adapted). In the model, the fraction of active excitatory neurons is qualitatively similar or even smaller for novel than for the adapted and onset stimuli (Figure 1- Figure Supplement 4A). Therefore, our model does not capture the dense novelty response as described in [Homann et al., 2017]. Why does this happen? Upon the presentation of a novel stimulus, inhibitory plasticity does not sufficiently increase inhibitory input into excitatory neurons to counteract the excess excitatory input. As a result, the high firing rates of excitatory neurons that are tuned to the novel stimulus strongly drive the entire inhibitory population. This in turn increases inhibition onto the entire excitatory population, including neurons not tuned to the novel stimulus. Since the firing rates of the excitatory neurons that are tuned to the adapted stimulus are lower (compared to the firing rates when presenting a novel stimulus), the adapted stimulus drives the entire inhibitory population less strongly reducing inhibition onto the entire excitatory population. Hence, the fraction of active excitatory neurons in the whole network is lower when presenting a novel stimulus compared to presenting an adapted stimulus (Figure 1—figure supplement 4A). Since the increase in inhibition seems to be responsible for the absence of a dense novelty response, we hypothesized that a dense novelty responses might result from a disinhibitory signal as explored in Figure 7. Indeed, in this scenario we observed an increase in the response density in our model (Figure 7C). The disinhibitory signal increases the fraction of active neurons as the strength of disinhibition increases (Figure 7E). Therefore, we hypothesize that the dense novelty response observed experimentally by [Homann et al., 2017] could be achieved by disinhibitory feedback.

We included a new Supplementary Figure (Figure 1—figure supplement 4) in which we show single neuron statistics (the fraction of active excitatory and inhibitory neurons for different stimuli) as well as the complete spike raster during an entire sequence block. We discuss this in lines 162, 389 and in the Discussion section in line 557 Furthermore, to be consistent throughout our study, we modified the measurement of density in Figure 7C, E (we also measure the fraction of active excitatory neurons).

4. It is unclear how duration of pre-training relates to plasticity and novelty responses (Figure 4B). Would you expect increased pretraining duration affect the novelty response behavior? If so, perhaps the duration of pre-training should be varied systematically? Furthermore, how does the number of pre-training stimuli affect the performance?

Varying the pretraining duration and the number of pretraining stimuli do not qualitatively change the novelty response and its properties (Figure 4—figure supplement 1): Novelty responses can be reliably detected across a large range of varied pretraining parameters. The novelty peak height increases with increased number of stimulus repetitions during pretraining and decreases with the number of different stimuli presented.

We followed the reviewer's suggestion and tested the effect of two parameters of the pretraining (the number of repetitions of each stimulus and the total number of stimuli presented during the pretraining phase) on the novelty response. In the standard repeated sequence paradigm (Figure 3, 4), a total of 65 stimuli were presented (5 x 3 repeated + 5 x 10 novel stimuli). Therefore, this is the baseline number of stimuli presented during pretraining. The number of repetitions refers to how often each of these 65 stimuli is presented. When testing the effect of the number of stimuli we varied the total number of stimuli presented 5 times each during pre-training. For example, 105 means that 40 additional subsets of neurons were stimulated during pretraining that are not part of the consecutive repeated sequence stimulation paradigm. We found that increasing the number of repetitions in the pretraining phase increases the novelty peak height before reaching a plateau at around 10 repetitions (Figure 4—figure supplement 1A), and increases the average inhibitory synaptic weights onto stimulus assemblies (Figure 4—figure supplement 1B). Increasing the number of stimuli decreases the novelty peak height (Figure 4—figure supplement 1C), but has little influence on the average inhibitory synaptic weights onto stimulus assemblies (Figure 4—figure supplement 1D). In summary, we found that while the pretraining parameters affect some aspects of the novelty response, they do not qualitatively impact our results. Even without a pretraining phase (zero number of repetitions), a novelty response occurs.

We included a new Figure 4—figure supplement 1 and provide a discussion of our findings on the pretraining parameters in line 285, in the Discussions section in line 536 and in the Methods section in line 781.

5. It is difficult to interpret the differences in adaptation between different repeated stimuli (Figure 4C bottom -> green vs blue lines) as compared to the differences between the "late" and "intermediate" time points (4D bottom). Results depicted in figure 4D suggest multiple timescales of adaptation. Are the differences in 4C, which seem to be of similar magnitude, meaningful?

The differences in the magnitude in Figure 4C are not meaningful and result from randomness in the model.

This aspect is indeed very important and we agree that it needs to be clarified. For each instantiation of the same model with different initial connectivity and assembly size, one would get a different ordering of the three traces corresponding to stimuli A, B and C in Figure 4C (bottom). Specifically, these differences are due to small differences in the initial random connectivity before the pretraining phase, which get amplified during the simulation due to plasticity. In particular, the strength of activation of different excitatory assemblies varies, leading to the variable synaptic connection strengths shown in Figure 4C (bottom).

In contrast, the differences in the inhibitory weights in Figure 4D (bottom) are meaningful and suggest multiple timescales of adaptation. By averaging out the differences due to randomness in Figure 4C (bottom), we find a clear increasing trend of the average inhibitory weights onto sequence 1 at later time points (Figure 4D, bottom). For another instantiation of the same model with different initial connectivity and assembly size, one would get exactly the same ordering of the three traces corresponding to early, intermediate and late in Figure 4D. This can for example also be seen in Figure 1—figure supplement 3D, where we test more complex stimuli (different in the pretraining and stimulated phases). The main reason for the increase in Figure 4D (bottom) follows from the strengthening of average inhibitory weights onto a specific sequence as the frequency of sequence presentation increases: at later time points the sequence has been presented more frequently than at earlier times, leading to the increase in average inhibitory weights (see also our answer to comment 7 of reviewer 1 and comment 1 of reviewer 3).

We now provide an additional explanation in the Results section (see line 280).

6. One of the primary findings of Natan et al., 2015 was the differential effect of optogenetic disinhibition of SOM vs PV interneurons on the SSA response. The differential effect is seen when disinhibiting the standards -1, +1, +2 etc. relative to the deviant. It would be interesting to see what the effect of disinhibition is on the network response (Figure 6), and how it relates to Natan's findings.

Disinhibiting the standards at certain post-deviant stimulus time-points (-1, 0, +1, +2, etc) in our model led to an equal increase of the firing rate response to standard tones (as with suppressing PVs in [Natan et al., 2015]); and a reduction in the firing rate change of the deviant stimulus compared to the standard stimuli (as with suppressing SOM in [Natan et al., 2015]).

Following the suggestion from the reviewer, we performed similar experiments as Figure 5 from [Natan et al., 2015]. In the experiments, whenever a tone was played, either the SOM or PV cells were optogenetically suppressed. Since our model only has a single class of inhibitory interneurons, the suppression was always applied to the entire inhibitory population. In particular, we applied disinhibition (suppressed the entire inhibitory population) during the standard (or repeated) stimulus A at time points -1, +1, +2, +3, +4 relative to the deviant stimulus (called the post-deviant stimulus number), and the deviant (or novel) stimulus B (post-deviant stimulus number 0) (Author response image 1). We found that suppressing the inhibitory population led to an equal increase in firing rates for all standard stimuli (post-deviant stimulus numbers -1, +1, +2, +3, +4) relative to the non-disinhibited case (Author response image 1). This agrees with the results of [Natan et al., 2015] when PV interneurons were suppressed, but not when SOM interneurons were suppressed, where the increase in responses to standard stimuli was dependent on the post-deviant stimulus number (compare Author response image 1 with Figure 5B from [Natan et al., 2015]). In addition, we observed a reduction in the firing rate of the deviant stimulus (post-deviant stimulus number 0) compared to the standard stimuli (post-deviant stimulus numbers -1, +1, +2, +3, +4) (compare red and grey bars in Author response image 1). This agrees with the results of [Natan et al., 2015] where SOM interneurons were suppressed (but note that in the experimental data, the deviant firing rate change was almost zero), but not when PV interneurons were suppressed. Therefore, our model captures some, but not all, aspects of the experimental data where multiple interneuron types were manipulated. Adding multiple interneuron subtypes (as in [Natan et al., 2015, Park and Geffen, 2020]) and possibly including different interneuron-specific plasticity rules [Agnes et al., 2020] is a promising line of future investigation.

We discuss this comparison in our manuscript line 651. In case the reviewers feel that the additional results from Author response image 1 would strengthen the manuscript and would make it easier to understand our study in the context of these previous experimental findings, we would be happy to include them in the manuscript.

**Author response image 1. respfig1:** Disinhibiting the standard and deviant stimulus leads to differential increase in responses to standard and deviant tones. A. Population firing rate of excitatory neurons in response to standard stimuli (gray) or to deviant stimuli (red) without disinhibition (dark colors) and with disinhibition (suppression of the total inhibitory population) (light colors). All responses are normalized to the response to the fourth non-disinhibited post-novelty stimulus of one instantiation of the model. B. Difference between firing rates with and without disinhibition from panel A in response to standard (gray) and deviant (red) stimuli. Error bars correspond to the standard deviation across three model instantiations.

7. The motivation for having different STDP learning rules for E->E and I->E connections is unclear, other than citation of previous work. Can you please explore in more detail the differences between these learning rules?

We now explore the motivation for using different learning rules for E-to-E and I-to-E connections in our manuscript. Our main motivation was to use biologically-inspired plasticity rules, which have important functional implications. For the plasticity of E-to-E and I-to-E connections multiple rules fulfil these requirements. Previous modelling studies have done extensive comparisons of the functional implications of multiple STDP rules. Hence, we did not implement different rules for our paradigm but discuss them in greater depth.

For excitatory-to-excitatory (E-to-E) synapses we used the triplet spike-timing-dependent plasticity (STDP) rule. More classically, the plasticity of E-to-E synapses has been argued to follow pair-based STDP, where the order and timing of pairs of spikes determines the induction of potentiation vs. depression [Bi and Poo, 1998]. Specifically, if a presynaptic spike comes before a postsynaptic spike the synapse is potentiated, while if a postsynaptic comes before a presynaptic spike then a synapse is depressed, as long as the timing between spikes is on the order of tens of milliseconds. However, such a pair-based STDP rule cannot explain plasticity where the frequency of pre- and postsynaptic spikes varies [Sjӧstrӧm et al., 2001]. To capture these data, the triplet STDP rule was proposed where a third spike modifies the amount of potentiation and depression evoked by pair-based STDP [Pfister and Gerstner, 2006]. Hence, triplets rather than pairs of spikes seem to be more appropriate as building blocks for synaptic plasticity for E-to-E synapses.

Functionally, the pair-based STDP rule cannot easily form neuronal assemblies; because pre-post spike pairs lead to potentiation and post-pre spike pairs lead to depression, the pair-based STDP rule leads to competition between reciprocal synapses, preventing the strengthening of bidirectional connections, and consequently self connected assemblies (but see e.g. [Babadi and Abbott, 2013]). Unlike the pair-based STDP rule, the triplet STDP rule supports the formation of bidirectional connections between neurons that experience correlated activity [Pfister and Gerstner, 2006, Gjorgjieva et al., 2011]. As a result, this rule can support the formation of self-connected assemblies in recurrent networks [Litwin-Kumar and Doiron, 2014, Zenke et al., 2015, Montangie et al., 2020]. Therefore, we used the triplet STDP rule for E-to-E connections to generate excitatory assemblies as a model for the different stimuli in a sequence (as shown in Figure 1—figure supplement 2C). Besides the triplet STDP rule, there exist also other E-to-E learning rules which allow for assembly formation. These include the voltage-based rule (see [Clopath et al., 2010] for details) and a calcium-based rule (see [Graupner and Brunel, 2010] for details). In a previous study, it has been shown that all three E-to-E learning rules allow for assembly formation (Figure 5 in [Litwin-Kumar and Doiron, 2014]).

For inhibitory-to-excitatory (I-to-E) synapses we used the inhibitory spike-timing-dependent plasticity (iSTDP) rule, initially suggested on theoretical grounds by [Vogels et al., 2011] for its ability to homeostatically stabilize firing rates in recurrent networks. According to this rule, the timing but not the order of pre- and postsynaptic spikes matters for the induction of synaptic plasticity. Hence, a pair of spikes that occurs within tens of milliseconds of each other can induce potentiation, and otherwise depression. This rule has been widely used in numerous computational models of recurrent networks to stabilize firing rate dynamics and balance excitation and inhibition ([Litwin-Kumar and Doiron, 2014, Zenke et al., 2015]; among others). More recently, this rule was also verified to operate in the auditory cortex of mice [D'amour and Froemke, 2015]. However, other inhibitory rules have also been measured experimentally, including classical Hebbian and anti-Hebbian (e.g. [Holmgren and Zilberter, 2001,Woodin et al., 2003, Haas et al., 2006]; for a review see [Hennequin et al., 2017]). Computational studies have started investigating the effect of such different inhibitory learning rules (see [Luz and Shamir, 2012, Kleberg et al., 2014]).

The generation of adapted and novelty responses in our model depends on a `negative feedback' mechanism of inhibitory plasticity (see also our response to comment 2 of reviewer 2). As long as inhibitory synapses potentiate in response to high excitatory firing rates, and decrease in response to low excitatory firing rates, the firing rates in response to repeated stimuli will decrease (i.e. adapt). Therefore, we expect that any inhibitory plasticity rule which incorporates a negative feedback mechanism will lead to the adaptation of responses to repeated stimuli. Another such candidate (besides the iSTDP rule we used) is a classical Hebbian inhibitory plasticity rule [Luz and Shamir, 2012]. We further explain the choice of inhibitory plasticity rule in response to comment 2 of reviewer 2, and comment 1 of reviewer 3.

We have included an additional supplementary figure to clarify our choice of the STDP functions for excitatory and inhibitory plasticity (Figure 1—figure supplement 1). To justify our choice of synaptic plasticity rules, we added text in the Results line 115, Methods (lines 709, 724 and 803) and Discussions (line 451).

8. An emerging theme from this paper and from other models such as Yarden and Nelken, 2017 and Park and Geffen, 2020, may be that the tonotopic organization of similarly tuned neurons helps facilitate adaptation. Tuned assemblies were a key feature of the models in these papers. Here, the tonotopic organization arises from the STDP rule here, and it would be interesting to discuss the relationship between tonotopic organization and plasticity.

In our model, tonotopy, or more generally topography, can emerge from the triplet excitatory STDP rule, but does not affect our modeling results and predictions.

Neuronal assemblies tuned to sensory stimuli are a key feature in our model. However, in its current implementation we do not have topographic organization of the assemblies { we refer to it as topographic here, because our model is sufficiently general to apply to the auditory system (tonotopy) as well as the visual system (retinotopy), which in fact was the main inspiration for our model (experiments in [Homann et al., 2017]). In our model, neuronal assemblies represent strongly connected subsets of excitatory neurons which form due to the functional properties of the triplet STDP shaping the plasticity between excitatory neurons. As we discussed previously in our answer to comment 7, the triplet STDP rule can strengthen connections between two neurons bidirectionally. Hence, the rule allows the formation of neuronal assemblies of neurons which receive similar inputs and experience correlated activity (see our previous work: [Gjorgjieva et al., 2011, Montangie et al., 2020]). We interpret these assemblies as being tuned to a given stimulus (e.g. orientated bar, or the frequency of a sound).

We could introduce topography (retinotopy or tonotopy) in our model by allowing our assemblies to overlap in a structured way. For example, we could define assembly 1 (coding for a given frequency) to have 50% overlap with assembly 2 (coding for a similar frequency), assembly 2 to have 50% overlap with assembly 3, etc. Qualitatively, introducing such topography would not affect our findings on the generation of adapted and novelty responses, and yet add additional complexity, so we decided to not include it.

We have expanded the discussion on assembly formation and how it could generate topography in our model (see lines 803 and 809). For the reasons mentioned above we have decided to not include topography in the model.

9. Seay et al., 2020, have considered plasticity rules in adaptation as a function of facilitation of specific inhibitory interneurons in SSA. It would be interesting to speculate whether and how this model and the learning rules relates to those published results.

Multiple differences in the model-set up between our model and the model in [Seay et al., 2020] make a direct comparison difficult. However, combining the mechanisms used in both studies is a promising future direction.

Using a computational model, [Seay et al., 2020] demonstrate that different experimentally measured short-term plasticity at the synapses from PV and SOM interneurons onto pyramidal neurons can account for diverse responses in the auditory cortex, from adapted to facilitated responses. In contrast in our study, we focus on long-term inhibitory plasticity and specifically on the generation of novelty responses. In addition, [Seay et al., 2020] study a feedforward circuit experiencing activity and plasticity on rather short timescales (e.g. 400 ms in the n+1 experiment), while we study a recurrent circuit operating on longer timescales (seconds to minutes, see Figure 4). These differences make it difficult to relate the two models (see also our answer to comment 3 of reviewer 3). However, in future work it would be interesting to combine the mechanism of inhibitory long-term plasticity that we implement, with the diverse short-term plasticity mechanisms from [Seay et al., 2020].

We added the reference on multiple occasions in our manuscript and discuss it on line 499.

Reviewer #2 (Recommendations for the authors):The units of the learning rate n of inhibitory plasticity on page 18 is listed in uF and only very briefly discussed? Since eta seems to be relatively important for the novelty detection, and determines e.g. how many trials a system will typically require to learn a stimulus (see also Figure 2) it should thus be discussed in more detail, possibly with an additional set of simulations that are complementary to those in Figure 2.

The timescale of inhibitory plasticity (η) is an important parameter in our model, which strongly influences the response amplitude and the decay time constant of the novelty response (Figure 4—figure supplement 2).

We followed the suggestion of the reviewer and systematically varied the learning rate η to see how it affects the novelty response. We found that the timescale of inhibitory plasticity needs to be `sufficiently fast' for the generation of the novelty response. For learning rates below η = 0:5 pF we no longer observe adaptation to repeated stimuli nor a novelty response (Figure 4—figure supplement 2A). The response amplitude of the novelty response increases (Figure 4—figure supplement 2B), while the decay time constant decreases (Figure 4—figure supplement 2C) with increasing inhibitory learning rate.

We have added new text in the Discussion under the subsections `Timescales of plasticity mechanisms' (line 509) and `Robustness of the model' (line 533) where we discuss the timescales of inhibitory plasticity. We also included a new supplementary figure (Figure 4—figure supplement 2) where we demonstrate the effect of varying η (see also our answer to comment 1 of reviewer 3), which we discuss in line 293 and mention in the Methods in line 739.

The inhibitory learning rule used here is stated to be confirmed on p. 15. This is arguable, and the jury may still be out. It may be worth discussing how other rules would perform?

Indeed, several inhibitory learning rules have been measured experimentally. We based our work on the iSTDP rule, originally proposed on a theoretical basis to homeostatically stabilize firing rate dynamics in recurrent networks [Vogels et al., 2011], and measured experimentally in the auditory cortex by [D'amour and Froemke, 2015]. This rule is symmetric in that the order of pre- and postsynaptic spikes does not matter for the induction of plasticity, only their timing. Other inhibitory rules have also been measured experimentally, including classical Hebbian and anti-Hebbian (e.g. [Holmgren and Zilberter, 2001,Woodin et al., 2003, Haas et al., 2006]; for a review see [Hennequin et al., 2017]). Furthermore, the learning rules seem to depend on the type of the interneuron [Udakis et al., 2020].

Computational studies have started investigating the effect of different inhibitory learning rules, albeit primarily in feedforward networks [Luz and Shamir, 2012, Kleberg et al., 2014]. The generation of adapted and novel responses in our model depends on a `negative feedback' mechanism of inhibitory plasticity. As long as inhibitory synapses potentiate in response to high excitatory firing rates, and decrease in response to low excitatory firing rates, the firing rates in response to repeated stimuli will decrease. Therefore, we expect that any inhibitory plasticity rule which incorporates a negative feedback mechanism would lead to adaptation of the responses to familiar stimuli. [Luz and Shamir, 2012] demonstrate that also a classical Hebbian inhibitory plasticity rule can implement negative feedback. See also our answer to related questions at comment 7 of reviewer 1 and comment 1 of reviewer 3.

We elaborate on this point in the Methods section, and mention other experimentally measured inhibitory learning rules (line 724) and their computational properties (line 451).

Reviewer #3 (Recommendations for the authors):1. The abstract is a bit cryptic: it states that "inhibitory synaptic plasticity readily generates novelty responses". Inhibitory plasticity is rather vague, it is necessary to read quite far into the results to figure out if the key is short-term synaptic inhibitory plasticity as in a number of models, or some form of long-term associative synaptic plasticity. The abstract should be more informative and explicitly state that the model relies on STDP of Inh->Ex synapse.

We modified the abstract as suggested by the reviewer and state that inhibitory spike-timing dependent plasticity is the underlying mechanism of adaptation in our model.

2. It would be helpful to provide a bit more information about the model architecture in Figure 1 and the start of the Results section so the reader does not immediately have to go to the Methods section (e.g., network size, ratio of E/I neurons, no STP).

We included further details of the model at the start of the Results section, see line 111.

3. In the Methods it would also be helpful to plot the STDP function for excitatory and inhibitory plasticity for spike pairs.

We added a new supplementary figure (Figure 1—figure supplement 1) which shows the STDP function for excitatory and inhibitory plasticity for spike pairs and mention it in lines 117 and 739. We did this for spike pairs of different frequencies because this is what makes the triplet STDP rule that we use very different from the more classical pair-based STDP rule.